



# Quantifying the buffering of oceanic oxygen isotopes at ancient midocean ridges

Yoshiki Kanzaki[1]

[1]Department of Earth and Planetary Sciences, University of California - Riverside, Riverside, CA 92521, USA

**Correspondence:** Yoshiki Kanzaki (kanzakiy@ucr.edu )

**Abstract.** To quantify the intensity of oceanic oxygen-isotope buffering through hydrothermal alteration of oceanic crust, a two-dimensional hydrothermal circulation model was coupled with a two-dimensional reactive transport model of oxygen isotopes. The coupled model calculates steady-state distributions of temperature, water flow and oxygen isotopes of solid rock and porewater given physicochemical conditions of oceanic crust alteration and seawater $\delta^{18}O$. Using the present-day seawater

$\delta^{18}O$ under plausible modern alteration conditions, the model yields $\delta^{18}O$ profiles for solid rock and porewater and fluxes of heat, water and $^{18}O$ that are consistent with modern observations, confirming the model's validity. The model was then run with different assumed seawater $\delta^{18}O$ values to evaluate oxygen isotopic buffering at the midocean ridges. The buffering intensity shown by the model is significantly weaker than previously assumed and, consistently, calculated $\delta^{18}O$ profiles of oceanic crust are relatively insensitive to seawater $\delta^{18}O$. These results are attributed to the fact that isotope exchange at shallow depths

does not reach equilibrium due to the relatively low temperatures, and $^{18}O$ supply via spreading solid rocks overwhelms that through water flow at deeper depths. Further model simulations under plausible alteration conditions during the Precambrian showed essentially the same results. Therefore, $\delta^{18}O$ records of ophiolites that are invariant at different Earth's ages can be explained by the relative insensitivity of oceanic rocks to seawater $\delta^{18}O$ and do not require constant seawater $\delta^{18}O$ through time.

## 1   Introduction

Hydrothermal alteration of oceanic crust at midocean ridges works as the dominant source/sink of several elements/isotopes in the ocean (e.g., Wolery and Sleep, 1976; Elderfield and Schultz, 1996). Notably, oxygen isotopes have been considered to be primarily controlled by isotope exchange at midocean ridges. The observation of oceanic crustal $\delta^{18}O$ has revealed that $^{18}O$

is added to and depleted from the oceanic crust at shallow and deeper depths, respectively, relative to the isotope amount of pristine crust. The close balance between the addition and removal of the heavy isotope, together with the huge oxygen supply from the mantle, has led to a hypothesis that the water-rock interactions at midocean ridges have buffered oceanic $\delta^{18}O$ at the present-day value (0 ‰ relative to standard mean ocean water, SMOW) throughout Earth's history (e.g., Muehlenbachs





and Clayton, 1976; Gregory and Taylor, 1981; Holland, 1984; Muehlenbachs, 1998). Age-invariant $\delta^{18}$O records of ophiolites
(ancient oceanic crust) have been argued to support the hypothesis (e.g., Holmden and Muehlenbachs, 1993).

In contrast to ophiolites, authigenic sedimentary rocks have shown secular $\delta^{18}$O increases with the Earth's age; sedimentary
rocks in the Precambrian are depleted in $^{18}$O by as much as 10 ‰ compared to those in the modern (e.g., Bindeman et al.,
2016). In theory, if constituting minerals of sedimentary rocks were formed in equilibrium with seawater and if the later
diagenetic/metamorphic modification of $\delta^{18}$O was negligible, $\delta^{18}$O records of authigenic sedimentary rocks can be utilized
to infer surface temperatures in the past, using temperature-dependent isotope fractionation factors and assuming the $\delta^{18}$O
value of seawater from which the constituting minerals formed. Indeed, several authors have suggested hot climates in the
Precambrian from $^{18}$O-depleted sedimentary records assuming the present-day seawater $\delta^{18}$O (e.g., 70–15 °C at 3.2–3.5 Ga
and 50–60 °C through the later Precambrian; Knauth and Lowe, 2003; Knauth, 2005). Such high temperatures are apparently
at odds with glacial records observed through the Precambrian (e.g., Catling and Kasting, 2017, Ch. 11). If one excludes hot
climates based on the glacial records, one must conclude that sedimentary $\delta^{18}$O records do not reflect seawater $\delta^{18}$O (e.g.,
instead reflecting diagenetic/metamorphic overprints) and/or that seawater $\delta^{18}$O has changed through time (e.g., Walker and
Lohmann, 1989). The latter conclusion has recently been supported by marine iron (oxyhydr)oxides, whose $\delta^{18}$O shows a
similar secular evolution to those in sedimentary rocks despite its little temperature dependence (Galili et al., 2019). On the
other hand, the secular change in seawawter $\delta^{18}$O apparently conflicts with the result derived from ophiolites: an invariant
seawater $\delta^{18}$O resulting from strong oceanic-$\delta^{18}$O buffering at midocean ridges. Overall, one cannot rely on sedimentary $\delta^{18}$O
to reconstruct ancient surface temperatures unless $\delta^{18}$O records of sedimentary rocks and ophiolites can be explained at the
same time.

Several hypotheses have been put forward to reconcile apparently conflicting invariant and variant $\delta^{18}$O records of respective
ophiolites and sedimentary rocks. Perry et al. (1978) pointed out that oceanic rocks in the Precambrian were more mafic than
45 those in the modern and this could have resulted in more intense low-temperature alteration of oceanic crust, removing more
$^{18}$O from seawater than today. Walker and Lohmann (1989) argued that the midocean ridges could have been above sea level
in shallower Precambrian oceans. Resultant subaerial water-rock interactions at low temperatures then could have removed
a large amount of $^{18}$O from the Precambrian oceans. Even if midocean ridges were below the sea level, lower pressures
on the seafloor in the shallow oceans could have brought less water into the oceanic crust, and water-rock interactions at
50 midocean ridges could have been dominated by those at low temperatures at shallower depths, again resulting in lower oceanic
$\delta^{18}$O in the Precambrian (Kasting et al., 2006). Absence of biogenic sediment cover during the Precambrian could have also
contributed to more significant low-temperature seafloor alteration and a lower seawater $\delta^{18}$O (e.g., Jaffrés et al., 2007). The
above arguments, however, give possible explanations only for low oceanic $\delta^{18}$O during the Precambrian; they do not explain
ophiolite $\delta^{18}$O records. Wallmann and colleagues have indicated that the invariant $\delta^{18}$O records of ophiolites can be explained
by decoupling ancient oceanic crust from the contemporaneous seawater $\delta^{18}$O (e.g., Wallmann, 2001; Jaffrés et al., 2007).
However, the simple box models considered by these authors cannot explain whether and/or how this decoupling could have
been made possible (Jaffrés et al., 2007).





To resolve the issue, a process-based approach that simulates oxygen isotope behavior during oceanic crust alteration is indispensable. Lécuyer and Allemand (1999) have developed an oxygen-isotope exchange model which utilizes prescribed
distributions of temperature and water/rock ratio within oceanic crust and an equation for half-closed systems (e.g., Gregory et al., 1989), and thus is not entirely process-based. Using this model, Lécuyer and Allemand (1999) concluded that oceanic $\delta^{18}$O cannot change from the present-day value by more than 2 ‰ due to the strong buffering exhibited by their modelled isotope-exchange at midocean ridges. This conclusion contrasts with that by Wallmann (2001), who adopted a box model to argue that the buffering must have been weak and that the oxygen-isotopic composition of the ocean has evolved through
the Phanerozoic. These conflicting conclusions from different models emphasize that a mechanistic understanding of oxygen isotope behavior during oceanic crust alteration is lacking.

The present study has been undertaken to present a process-based model to simulate oxygen isotope behavior during hydrothermal alteration of oceanic crust and its application to the Precambrian. A two-dimensional (2D) hydrothermal circulation model presented in the literature is combined with a 2D reactive transport model of oxygen isotopes. The coupled model can
thus yield 2D distributions of solid-rock and porewater $\delta^{18}$O reflecting alteration conditions including seawater $\delta^{18}$O and hydrothermal fluid circulation. After confirming the model's validity by comparing the model results that assume the present-day seawater $\delta^{18}$O with modern observations, we examine the intensity of oceanic-$\delta^{18}$O buffering by hydrothermal alteration of oceanic crust at midocean ridges by changing seawater $\delta^{18}$O. The buffering quantification is then conducted under different physicochemical conditions that could have been the case during the Precambrian, to give insights into how oceanic $\delta^{18}$O could
have been affected by water-rock interactions at midocean ridges during the Earth's early eons.

## 2 Methods

### 2.1 Hydrothermal circulation model

Hydrothermal flow circulating around midocean ridges is simulated two-dimensionally based on conservations of energy, mass and momentum (Steefel and Lasaga, 1994; Cherkaoui et al., 2003; Iyer et al., 2010). Assuming steady state, the mass
conservation of fluid is represented by

$$\boldsymbol{\nabla} \cdot \mathbf{q} = 0, \tag{1}$$

where $\boldsymbol{\nabla}$ is the vector differential operator ($\boldsymbol{\nabla} = (\partial/\partial x, \partial/\partial y)$) and $\mathbf{q}$ is the mass flux vector (kg m$^{-2}$ yr$^{-1}$). Conservation of momentum is realized by adopting Darcy's law (e.g., Steefel and Lasaga, 1994):

$$\mathbf{q} = -\frac{k}{\nu}\left(\boldsymbol{\nabla} P - \rho_\mathrm{f} \mathbf{g}\right), \tag{2}$$

where $k$ is the permeability of oceanic bulk rock (m$^2$), $\nu$ is the kinematic viscosity of water (m$^2$ yr$^{-1}$), $P$ is the fluid pressure (Pa), $\rho_\mathrm{f}$ is the fluid density (kg m$^{-3}$) and $\mathbf{g}$ is the gravity vector given by $\mathbf{g} = (0, -g)$ where $g$ represents the acceleration by gravity (m yr$^{-2}$) (e.g., Steefel and Lasaga, 1994). The energy conservation is represented by

$$\left\{\phi\rho_\mathrm{f} c_\mathrm{p}^\mathrm{f} + (1-\phi)\rho_\mathrm{m} c_\mathrm{p}^\mathrm{m}\right\}\frac{\partial T}{\partial t} = \boldsymbol{\nabla} \cdot \left(-\mathbf{q}c_\mathrm{p}^\mathrm{f} T + \kappa \boldsymbol{\nabla} T\right), \tag{3}$$





where $t$ is time (yr), $\phi$ is the porosity, $\rho_\mathrm{m}$ is the density of oceanic rock (kg m$^{-3}$), $c_\mathrm{p}^\mathrm{f}$ and $c_\mathrm{p}^\mathrm{m}$ are the specific heat capacity of water and oceanic rock, respectively (J kg$^{-1}$ K$^{-1}$), $T$ is the temperature (K) and $\kappa$ is the thermal conductivity of oceanic rock (J yr$^{-1}$ m$^{-1}$ K$^{-1}$). The thermodynamic and transport properties of water ($c_\mathrm{p}^\mathrm{f}$, $\rho_\mathrm{f}$ and $\nu$) are obtained through a FORTRAN90 library STEAM which is based on the NBS steam table package (Meyer et al., 1983; Haar et al., 1984). As in Iyer et al. (2010), we assume pure water for hydrothermal fluid. The petrophysical parameters except for the permeability (i.e., $\kappa$, $\phi$, $\rho_\mathrm{m}$ and $c_\mathrm{p}^\mathrm{m}$) are assumed to be the same as those in Iyer et al. (2010). We assume that $\log k$ exponentially decreases from $-11.8$ to $-16.8$ with a scale length of 300 m for the e-fold decrease, which is consistent with observations (Fisher, 1998; Supplementary material). Following Cherkaoui et al. (2003), oceanic rocks below 6 km depth from the ocean/crust interface and with temperatures above the critical temperature for rock cracking (600 °C) are impermeable. As long as the model is consistent with observations, changing the assumptions about permeability does not affect the general conclusions given in the present study (Supplementary material). See Table 1 for the definitions and values of the parameters used in the present study.

A finite difference approach is taken to solve Eqs. (1)–(3) for **q**, $P$ and $T$ using the second-order central differencing scheme for the second-order differential terms and the first-order upwind and forward differencing schemes for the first-order spatial and temporal differential terms, respectively (e.g., Steefel and Lasaga, 1994). The calculation procedure follows that by Iyer et al. (2010). First, we obtain $P$ by solving Eqs. (1) and (2). Then, **q** is obtained from the calculated $P$ and Eq. (2). Finally, the calculated **q** is used in Eq. (3) which is solved to obtain $T$ for the same time step. The calculated $P$ and $T$ are used to update the thermodynamic and transport properties of water for the next calculation step. The above procedure is repeated with a time step of $3 \times 10^4$ yr until steady states are reached, which is accomplished within $10^3$ time steps (cf. Cherkaoui et al., 2003). The model grid extends to 12 km depth from the ocean/crust interface and to 30 km distance from the ridge axis. The calculation domain ($12 \times 30$ km$^2$) is divided into a $320 \times 200$ irregular grid, with the grid-cell size horizontally increasing from the ridge axis (1.1 to 330 m) and vertically from the ocean/crust interface (0.17 to 82 m).

The boundary conditions adopted in the present study follow those by Iyer et al. (2010) and/or Cherkaoui et al. (2003). Pressure at the ocean/crust interface is assumed as constant at 25 MPa, corresponding to the assumption of 2.5 km water depth. This assumption is modified when we explore the Precambrian hydrothermal circulation in Section 3.3. The right and bottom sides of the calculation domain are assumed to be insulating. The ocean above the top boundary is assumed to have a constant temperature of 2 °C. The ridge axis on the left-hand boundary is assumed to supply a boundary heat flux $J_\mathrm{b}$ (J m$^{-2}$ yr$^{-1}$):

$$J_\mathrm{b} = \rho_\mathrm{m} w \left( T_\mathrm{m} - T \right) c_\mathrm{p}^\mathrm{m}, \tag{4}$$

where $w$ is the spreading rate (m yr$^{-1}$) and $T_\mathrm{m}$ is the temperature of the intrusion (1200 °C). As a standard value, we assume $3 \times 10^{-2}$ m yr$^{-1}$ for $w$, and this assumption is again changed in Section 3.3 where we consider the Precambrian hydrothermal circulation. Free flow is allowed at the top boundary, and the other boundaries are assumed to be impermeable (Iyer et al., 2010). The calculation domain is wide enough that making the right-hand boundary permeable (cf. Cherkaoui et al., 2003) changes the results only negligibly.



## 2.2 Reactive transport model of oxygen isotopes

Oxygen isotopes of solid rock are assumed to be transported by the spreading of oceanic rocks, while those of porewater are transported by water flow and molecular diffusion (cf. Norton and Taylor, 1979; Lécuyer and Allemand, 1999). At the same time, isotope exchange reactions transfer $^{18}$O between the two phases (e.g., Norton and Taylor, 1979; Lécuyer and Allemand, 1999). Then, from the mass conservation of $^{18}$O in the two phases, the time rates of change of the $^{18}$O/total O mole ratios of solid rock and porewater ($F_r$ and $F_p$, respectively) can be represented by

$$(1 - \phi)\,\rho_m m_s \frac{\partial F_r}{\partial t} = -(1 - \phi)\,\rho_m m_s w \frac{\partial F_r}{\partial x} - \rho_b m_s m_f k_{ex} \left\{ F_r \left(1 - F_p\right) - \alpha \left(1 - F_r\right) F_p \right\}, \text{ and} \tag{5}$$

$$\phi \rho_f m_f \frac{\partial F_p}{\partial t} = \boldsymbol{\nabla} \cdot \left(-m_f \mathbf{q} F_p + \phi \rho_f m_f D \boldsymbol{\nabla} F_p\right) + \rho_b m_s m_f k_{ex} \left\{ F_r \left(1 - F_p\right) - \alpha \left(1 - F_r\right) F_p \right\}, \tag{6}$$

where $m_s$ and $m_f$ are the oxygen concentrations of solid rock and porewater, respectively (mol kg$^{-1}$), $\rho_b$ is the density of bulk rock given by $\rho_b = \phi \rho_f + (1 - \phi)\rho_m$, $k_{ex}$ is the rate constant for oxygen isotope exchange (mol$^{-1}$ kg yr$^{-1}$), $\alpha$ is the oxygen isotope fractionation factor and $D$ is the effective diffusion coefficient for $^{18}$O (m$^2$ yr$^{-1}$), determined by molecular diffusion and hydrodynamic dispersion. The first and second terms on the right-hand sides of Eqs. (5) and (6) represent oxygen isotope transport and oxygen isotope exchange, respectively. The kinetic expression of oxygen isotope exchange in Eqs. (5) and (6) is formulated based on Cole et al. (1983, 1987) and the rate constant is given by an Arrhenius equation:

$$k_{ex} = 10^{-8.5} \exp \left\{ -\frac{E}{R_g} \left( \frac{1}{T} - \frac{1}{278} \right) \right\}, \tag{7}$$

where $E$ is the apparent activation energy ($5 \times 10^4$ J mol$^{-1}$) and $R_g$ is the gas constant (8.314 J mol$^{-1}$ K$^{-1}$). The value of $10^{-8.5}$ mol$^{-1}$ kg yr$^{-1}$ at reference temperature 278 K (5 °C) is comparable to the $10^{-7.2}$–$10^{-6.6}$ mol$^{-1}$ kg yr$^{-1}$ at 5 °C extrapolated from $10^{-2.1}$–$10^{-1.5}$ mol m$^{-2}$ yr$^{-1}$ at 300 °C (Cole et al., 1987) with $E = 5 \times 10^4$ J mol$^{-1}$ (cf. Cole et al., 1987) and $10^3$ m$^2$ kg$^{-1}$ specific surface area of marine basalt (cf. Nielsen and Fisk, 2010), given that the reaction rate in the field is generally slower than in the laboratory by a factor of up to $10^3$ (e.g., Wallmann et al., 2008). Note that the general results and conclusions in the present study are not affected by changes in the reference $k_{ex}$ value within a plausible range (Supplementary material). The oxygen isotope fractionation factor for andesite by Zhao and Zheng (2003) is adopted in the present study:

$$10^3 \ln \alpha = \beta \left( \frac{6.673 \times 10^6}{T^2} + \frac{10.398 \times 10^3}{T} - 4.78 \right) \exp \left( \frac{1 - \beta}{R_g T} \right) - \frac{2.194 \times 10^6}{T^2} - \frac{15.163 \times 10^3}{T} + 4.72 + 1.767 \left( 2\beta - 1 \right), \tag{8}$$

where $\beta = 0.876$ (Zhao and Zheng, 2003), because Eq. (8) with $\beta = 0.876$ yields comparable $\alpha$ values to experimental results for basalt by Cole et al. (1987) while applicable over the wide range of temperature considered for the present study (0–1200 °C). The effective diffusion coefficient $D$ considers both molecular diffusion and hydrodynamic dispersion; the former is obtained from the modified Stokes-Einstein relation for H$_2$$^{16}$O diffusion by Krynicki et al. (1978) and the isotope effect from Harris and Woolf (1980) and a homogeneous dispersivity of 10 m is assumed for the latter (Frind, 1982; Gelhar et al., 1992):

$$D = \tau 6.9 \times 10^{-15} \frac{T}{\rho_f \nu} \sqrt{\frac{18}{20}} + 10 \frac{|\mathbf{q}|}{\phi \rho_f}. \tag{9}$$





The first term in the right-hand side of Eq. (9) represents the molecular diffusion including the tortuosity factor $\tau = \phi^{1.4}$ (Aachib et al., 2004), while the second term the hydrodynamic dispersion. The petrophysical parameters are assumed to be the same as those in the hydrothermal circulation model (Section 2.1, Table 1). The thermodynamic and transport properties of water are obtained through the hydrothermal circulation model (Section 2.1).

The steady-state values of $F_r$ and $F_p$ are obtained by solving $\partial F_r / \partial t = \partial F_p / \partial t = 0$ in Eqs. (5) and (6). The intrusion on the left-hand boundary is assumed to have a $^{18}$O/total O mole ratio of fresh crust ($F_m$) that corresponds to 5.7 ‰ relative to SMOW (e.g., Holmden and Muehlenbachs, 1993), and the ocean above the top boundary is assumed to have a constant $^{18}$O/total O mole ratio of seawater ($F_{sw}$). The other boundaries are impermeable for $^{18}$O fluxes via water. A finite difference method is used for equation discretization (first-order upwind and second-order central differencing schemes for the first-order and second-order spatial differential terms, respectively) and Newton's method is adopted to solve the difference equations.

The calculation is conducted on the grid described earlier for the hydrothermal circulation model (Section 2.1). The $^{18}$O mole ratios are all reported in $\delta$ notation relative to SMOW, using $2.0052 \times 10^{-3}$ as the $^{18}$O/$^{16}$O mole ratio of SMOW (Fry, 2006).

## 3 Results

### 3.1 Application to the present day and model validation

The calculated flow geometry and temperature distribution (Fig. 1; note that Figs. 1c and d are plots of Figs. 1a and b, respectively, on logarithmic scales) are similar to those in previous studies (e.g., Cherkaoui et al., 2003). The 2D model results can be converted to associated mass and heat fluxes, assuming $10^8$ m ridge length (cf. Wolery and Sleep, 1976). The total heat flux from the system is $0.74 \times 10^{12}$ W, comparable to the observed cumulative heat flux within 1 Ma (corresponding to 30 km with $3 \times 10^{-2}$ m yr$^{-1}$ spreading rate) from the ridge axis, $0.4(\pm 0.3) \times 10^{12}$ W (Stein and Stein, 1994). The total mass of water

exchange is $1.2 \times 10^{13}$ kg yr$^{-1}$, falling within the constrained range by Elderfield and Schultz (1996) ($3 \times 10^{12}$ to $1.6 \times 10^{14}$ kg yr$^{-1}$) and comparable to the prediction from other hydrothermal circulation models (e.g., Norton and Knight, 1977; Fehn et al., 1983; Cherkaoui et al., 2003). The calculated 2D distributions of solid-rock and porewater $\delta^{18}$O (Figs. 2a, 2d, 3a and 3d; note that Fig 3 is a plot of Fig. 2 on logarithmic scales) are consistent with observations, especially at 30 km (1 Ma) from the ridge axis (black curves in Fig. 4; see Section 3.2 for paler curves). Where flow rate and temperature are highest near the ridge

axis and close to the ocean/crust interface (at $< \sim 200$ m depths), porewater $\delta^{18}$O is in the range between 0 and 2 ‰ (Fig. 3d), which is comparable to the observed $\delta^{18}$O ranges for high-temperature hydrothermal fluids, e.g., 0.49–2.3 ‰ by Jean-Baptiste et al. (1997) and 0.3–1.4 ‰ by James et al. (2014). Away from the ridge axis, porewater $\delta^{18}$O at the ocean/crust interface is equivalent to the seawater value, but drops down to $-4$ to $-14$ ‰ within $\sim 200$ m depths, again not inconsistent with those receiving small water exchange, down to $-8$ ‰ within 500 m depths (e.g., Lawrence and Gieskes, 1981). Solid rock $\delta^{18}$O

at 1 Ma (i.e., 30 km from the ridge axis) is characterized by high ($< 16$ ‰) and low ($> 3$ ‰) values at shallow ($\geq \sim 2$ km) and deeper ($\leq \sim 2$ km) depths, respectively, consistent with observations of modern oceanic crust (e.g., Alt et al., 1986) and ophiolites (e.g., Gregory and Taylor, 1981; crosses in Fig. 4). The $^{18}$O fluxes to the ocean from high- and low-temperature alteration are $3.0 \times 10^9$ and $-2.8 \times 10^9$ mol yr$^{-1}$, respectively, well balanced, resulting in a net $^{18}$O flux of $0.2 \times 10^9$ mol



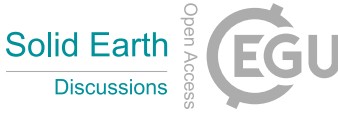

$\mathrm{yr}^{-1}$, consistent with the suggestion of zero net-flux by Gregory and Taylor (1981). Individual $^{18}$O-flux values are also comparable to those suggested in the previous studies. As examples, $^{18}$O fluxes through high-temperature alteration have been suggested to be $4.5 \times 10^9$ (Holland, 1984), $3.2 \times 10^9$ (Muehlenbachs, 1998) and $2.8 \times 10^9$ $\mathrm{mol\ yr}^{-1}$ (Wallmann, 2001), and those through low-temperature alteration $< -2.3 \times 10^9$ $\mathrm{mol\ yr}^{-1}$ (Lawrence and Gieskes, 1981), $-1.1 \times 10^9$ (Holland, 1984), $-1.0 \times 10^9$ (Muehlenbachs, 1998) and $-0.9 \times 10^9$ $\mathrm{mol\ yr}^{-1}$ (Wallmann, 2001). The consistency between the model calculation and observations described above supports the validity of the model.

## 3.2 Evaluation of oceanic-$\delta^{18}$O buffering capacity

As both the source and sink of oceanic $^{18}$O, hydrothermal systems can buffer oceanic $\delta^{18}$O but the strength of the buffering depends on the sensitivity of isotope exchange between oceanic crust and porewater to seawater $\delta^{18}$O (cf. Wallmann, 2001). As an extreme example, if the oxygen isotope fractionation between solid rock and porewater is independent of seawater $\delta^{18}$O, there should not be any feedbacks from the hydrothermal systems on changes in seawater $\delta^{18}$O, i.e., no oceanic-$\delta^{18}$O buffering. The previous studies that support the strong oceanic-$\delta^{18}$O buffering at midocean ridges assume linear relationships between oxygen isotope fractionation and seawater $\delta^{18}$O (e.g., Gregory and Taylor, 1981; Holland, 1984; Muehlenbachs, 1998). This assumption regarding the sensitivity to seawater $\delta^{18}$O is difficult to confirm through observations of geological records because seawater $\delta^{18}$O is not known. We can examine the response of rocks to seawater $\delta^{18}$O with the present model by adopting different values for seawater $\delta^{18}$O. We can then measure the buffering capacity of the system by plotting net $^{18}$O flux against seawater $\delta^{18}$O and calculating the slope value, i.e., $\partial(\text{net }^{18}\text{O flux})/\partial(\text{seawater }\delta^{18}\text{O})$, as in, e.g., Muehlenbachs and Clayton (1976). For example, a hydrothermal system with a large negative slope value should exhibit a strong buffering of oceanic $\delta^{18}$O, because a slight change in seawater $\delta^{18}$O makes the system introduce a large net $^{18}$O flux to the ocean that restores the change in seawater $\delta^{18}$O (e.g., Muehlenbachs and Clayton, 1976; Muehlenbachs, 1998).

Most features of 2D distributions of solid-rock and porewater $\delta^{18}$O are not significantly affected by decreasing seawater $\delta^{18}$O from 0 to $-12$ ‰ (Figs. 2–4). Although the porewaters close to the ocean have $\delta^{18}$O compositions close to those of seawater (e.g., Fig. 3), $\delta^{18}$O signatures of solid rocks are not linearly proportional to seawater $\delta^{18}$O (e.g., Fig. 4). The relative insensitivity at shallow depths ($\leq \sim 2$ km; Figs. 2–4) can be explained by the kinetics of oxygen isotope exchange. The distance from isotope exchange equilibrium can be measured by $\Omega = F_\mathrm{r}(1 - F_\mathrm{p})/\{\alpha(1 - F_\mathrm{r})F_\mathrm{p}\}$ (Eqs. (5) and (6); Figs. 5b and d; note that Figs. 5c and d are plots of Figs. 5a and b, respectively, on logarithmic scales), and the shallow regions show non-equilibrium states ($\Omega \neq 1$) because of their relatively low temperatures (Figs. 1b and d). Therefore, despite the approximate isotopic equivalence between porewater and seawater (Figs. 2 and 3), solid rocks do not directly reflect seawater $\delta^{18}$O in their oxygen-isotopic compositions. In deeper sections of oceanic crust, on the other hand, solid rocks attain isotope exchange equilibrium with porewater (e.g., Figs. 5b and d). These deep regions, however, receive less water exchange than the shallow regions, as can be seen from the distribution of local water/rock oxygen-mole ratio $\eta$ ($\equiv m_\mathrm{f}|\mathbf{q}|/\{(1 - \phi)\rho_\mathrm{m} m_\mathrm{s} w\}$) (Figs. 5a and c). Accordingly, deep oceanic rocks and porewaters are oxygen-isotopically buffered by spreading solid-rocks rather than by seawater, and $\delta^{18}$O values of deep porewaters are almost independent of seawater $\delta^{18}$O (Figs. 2d–f and 3d–f). Consequently, despite the isotope exchange equilibrium, deep oceanic rocks are almost completely insensitive to seawater $\delta^{18}$O. The combi-



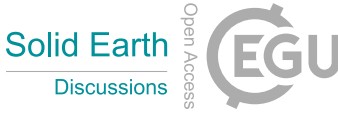

nation of the above two factors (isotope exchange kinetics and $^{18}$O supply from spreading solid-rocks) explains the insensitivity of midocean ridge systems to seawater $\delta^{18}$O (Figs. 2–4).

Consistent with the insensitivity of oceanic rocks to seawater $\delta^{18}$O (Figs. 2–4), the net $^{18}$O flux from the midocean ridge systems is a very weak function of seawater $\delta^{18}$O (Fig. 6). Compared to the large buffering capacity suggested in the previous studies, which can be recognized from large negative slopes in Fig. 6, the intensity of oceanic-$\delta^{18}$O buffering exhibited by the present model is weak; the magnitude of slope ($-0.4 \times 10^9$ mol yr$^{-1}$ ‰$^{-1}$) is smaller than those previously assumed by a factor of up to >7. The reason why the previous studies suggested strong buffering is because these studies assume

isotope exchange equilibrium between porewaters and oceanic rocks and equivalence between porewater and seawater. These assumptions are generally not applicable to midocean ridge systems, as can be anticipated from Figs. 2–5. The exception to the previous studies is Wallmann (2001), which shows relatively small negative slope value (Fig. 6). Note, however, that Wallmann (2001) simulated weak buffering in a different way, i.e., by adopting low and temperature-independent rate constants for oxygen isotope exchange.

### 230  3.3   Application to the Precambrian

As the tectonics of the Earth likely evolved through time, the midocean ridge systems in the Precambrian could have been quite different from those in the present day. Indeed, differences in ocean volume and the spreading rate of oceanic crust have been discussed (e.g., Kasting et al., 2006; Korenaga et al., 2017). Here, we examine the system responses to changes in the spreading rate and ocean volume regarding oceanic-$\delta^{18}$O buffering.

Increasing the spreading rate from $1 \times 10^{-2}$ to $30 \times 10^{-2}$ m yr$^{-1}$ results in increases in the heat flux ($0.50 \times 10^{12}$, $0.99 \times 10^{12}$ and $1.25 \times 10^{12}$ W at $1 \times 10^{-2}$, $9 \times 10^{-2}$ and $30 \times 10^{-2}$ m yr$^{-1}$, respectively; Eq. (4)) and associated water exchange between the crust and ocean ($2.4 \times 10^{12}$, $4.9 \times 10^{13}$ and $8.5 \times 10^{13}$ kg yr$^{-1}$ at $1 \times 10^{-2}$, $9 \times 10^{-2}$ and $30 \times 10^{-2}$ m yr$^{-1}$, respectively) (Figs. 7a–f), not inconsistent with modern observations (e.g., Baker et al., 1996; Bach and Humphris, 1999). On the other hand, differences in temperature and water-flow distributions caused by those in ocean depth (from 1 to 5 km) are relatively minor

($0.74 \times 10^{12}$ W and ($1.2$–$1.3$)$\times 10^{13}$ kg yr$^{-1}$; Figs. 7g–j). The distributions of solid-rock and porewater $\delta^{18}$O predicted under these different alteration conditions are correspondingly modified but show features common to those shown in the previous sections with the standard parameterization. Porewater $\delta^{18}$O is close to seawater $\delta^{18}$O only along the ocean/crust interface and becomes quite different from seawater $\delta^{18}$O at deep depths (Supplementary material). Solid rock $\delta^{18}$O is relatively insensitive to seawater $\delta^{18}$O in general, showing relative addition and removal of $^{18}$O at shallow and deeper depths, respectively (Fig.

8). The buffering capacity exhibited by models that reflect changes in the spreading rate and ocean depth is different from that in the standard case (Fig. 9), but still weaker than any of those assumed in the previous studies (cf. Fig. 6). Note that in the previous studies, the buffering capacity is assumed to increase linearly with the spreading rate (Gregory and Taylor, 1981; Holland, 1984; Muehlenbachs, 1998; Wallmann, 2001) and thus the slope values in Fig. 9 need be compared with those in the literature (Fig. 6) multiplied by a factor that accounts for changes in the spreading rate. As an example, the slope value of

$-1.9 \times 10^9$ mol yr$^{-1}$ ‰$^{-1}$ with a spreading rate $30 \times 10^{-2}$ m yr$^{-1}$ in Fig. 9 should be compared with $-6 \times 10^9$ (Wallmann, 2001), $-(14$–$27) \times 10^9$ (Gregory and Taylor, 1981), $-24 \times 10^9$ (Muehlenbachs, 1998) and $-31 \times 10^9$ mol yr$^{-1}$ ‰$^{-1}$ (Holland,



1984) (cf. Fig. 6); thus the buffering with the presernt model assuming $30 \times 10^{-2}$ m yr$^{-1}$ of sprading rate is much waeker than those assumed in the previous study. More details of changes in the system behavior are described below.

When spreading is slower, the resultant flow geometry shows less intense but more homogenized mixing (Fig. 7a) and the changes of crustal $\delta^{18}$O from the pristine $\delta^{18}$O value (5.7 ‰) become larger over the whole crust (Fig. 8a). The sensitivity of solid-rock to seawater $\delta^{18}$O is small (Fig. 8a) because of low water/rock ratios resulting from the less intense water mixing (Fig. 7a) overwhelming the smaller supply of $^{18}$O via spreading rocks. The small sensitivity combined with the lower O supply from the mantle makes the buffering intensity weaker than that in the standard case (Fig. 9). With the spreading rate high, on the other hand, hydrothermal flows are more intense but more localized close to the ocean and the ridge axis (Figs. 7c and e). Accordingly, $^{18}$O depletion from solid rocks near the ridge axis at relatively high temperatures and water/rock ratios is not recovered even after continued reactions at low temperatures, making the system more sensitive to seawater $\delta^{18}$O (Figs. 8b and c). With the higher rock sensitivity to seawater $\delta^{18}$O, combined with the larger O supply from the mantle, the buffering capacity is larger with the higher spreading rate than with the standard spreading rate (Fig. 9). Overall, the system as a whole exhibits non-linear and reduced sensitivity to changes in the spreading rate as recognized from the relationship between the spreading rate and buffering capacity in Fig. 9 (cf. Gregory and Taylor, 1981; Holland, 1984; Muehlenbachs, 1998; Wallmann, 2001).

When assuming shallow ocean and thus water depth (1 km), water mixing becomes slightly stronger and localized slightly closer to the ocean (Fig. 7g) (cf. Kasting et al., 2006). Otherwise, changes are negligible compared to our standard case and the resultant oxygen-isotope behavior is relatively similar to that in the standard case (Figs. 4 and 8d). The situation is almost the same when we consider deeper ocean with a water depth 5 km, except that the mixing localization shifts slightly deeper and the mixing is slightly weaker compared to shallower ocean cases (Fig. 7i), and correspondingly $\delta^{18}$O distributions are slightly modified (Fig. 8e). In both shallow- and deep-ocean cases, the oceanic-$\delta^{18}$O buffering becomes only slightly weaker than in the standard case (Fig. 9), attributable to slight changes in the flow geometry and intensity. Because of the slight shifts in water mixing localization, isotope exchange at low temperature is relatively enhanced in shallow oceans (Fig. 9), not inconsistent with the prediction by Kasting et al. (2006). On the other hand, the magnitude of the ocean-depth effect is much smaller than anticipated by Kasting et al. (2006), probably because of the settings in the hydrothermal circulation model (Section 2.1) where permeability distribution dominantly controls the amount of water exchange (e.g., Cherkaoui et al., 2003; Supplementary material).

## 4 Discussion

### 4.1 Interpretation of ophiolites

Simulations conducted in the present study suggest that oceanic rocks are not significantly affected by changes in seawater $\delta^{18}$O under any plausible alteration conditions (Figs. 4 and 8). Reported $\delta^{18}$O values of ophiolites and/or oceanic crust range from $\sim 1$ to 16 ‰ (dashed lines in Fig. 8). By comparison, the simulated solid-rock $\delta^{18}$O values fall within this range at $\geq \sim -8$ ‰ seawater $\delta^{18}$O with $3 \times 10^{-2}$ m yr$^{-1}$ of spreading rate (Figs. 4, 8d and 8e), at $\leq \sim -10$ ‰ seawater $\delta^{18}$O with





$1 \times 10^{-2}$ m yr$^{-1}$ of spreading rate (Fig. 8a), at $\geq\sim -2$ ‰ seawater $\delta^{18}$O with $9 \times 10^{-2}$ m yr$^{-1}$ of spreading rate (Fig. 8b) and at $\geq\sim -2$ ‰ seawater $\delta^{18}$O with $30 \times 10^{-2}$ m yr$^{-1}$ of spreading rate (Fig. 8c). Accordingly, we conclude that the constant seawater $\delta^{18}$O at 0 ‰ is neither a necessary nor a sufficient condition for explaining the relatively invariant $\delta^{18}$O records of ophiolites. As ophiolite $\delta^{18}$O profiles can be affected more by alteration conditions (e.g., the spreading rate and permeability; Section 3.3 and Supplementary material) with the control by seawater $\delta^{18}$O remaining relatively weak, feedbacks between the alteration parameters (e.g., spreading rate and permeability) could be more important in reproducing ophiolite records. Therefore, ophiolites may be interpreted to indicate the insensitivity of oceanic rocks to seawater $\delta^{18}$O, realized by the feedbacks between alteration parameters, and not necessarily a constant seawater $\delta^{18}$O. The weak buffering (Figs. 6 and 9) accompanying the partial decoupling between the oceanic crust and seawater $\delta^{18}$O (e.g., Fig. 8) shows that this interpretation is more plausible than the constant seawater $\delta^{18}$O.

## 4.2 Controls of $\delta^{18}$O in the Precambrian oceans

Although the buffering of oxygen isotopes through hydrothermal alteration of oceanic rocks is weaker than previously assumed under any alteration conditions (Section 3.3), it could have been relatively strong when the spreading rate is $\geq\sim 30 \times 10^{-2}$ m yr$^{-1}$ (Figs. 6 and 9). Such high-spreading rate conditions could have been possible only during the earliest period of Earth's history ($> 3.5$ Ga; Phipps Morgan, 1998) or even impossible according to Korenaga (2006). Excluding this earliest period, the buffering must have been weak despite the uncertainties in alteration conditions (Figs. 6 and 9). The weak buffering must have allowed variations of seawater $\delta^{18}$O through other surficial processes that exchange $^{18}$O with seawater, most likely through continental weathering (e.g., Walker and Lohmann, 1989). As surface environments likely have significantly changed through the eons (cf. Introduction), $^{18}$O fluxes from continental weathering could have correspondingly varied throughout Earth's history.

Previous studies examining oceanic-$\delta^{18}$O evolution on geological time scales have utilized box-modeling approaches to account for oxygen isotope exchange from both continental weathering and hydrothermal alteration of oceanic crust (Godderis and Veizer, 2000; Goddéris et al., 2001; Wallmann, 2001; Kasting et al., 2006; Jaffrés et al., 2007). Among them, the studies that assume strong buffering at midocean ridges have had difficulty in modifying oceanic $^{18}$O budget through other surficial processes including continental weathering (Godderis and Veizer, 2000; Goddéris et al., 2001). Other studies that assume weak buffering instead have shown the possibility of significant oceanic-$\delta^{18}$O changes through modification of simplified continental-weathering parameters with the Earth's age (Wallmann, 2001; Kasting et al., 2006; Jaffrés et al., 2007). Revisiting theses previous box-model studies with the weak buffering suggested here (e.g., Figs. 6 and 9) and constructing a process-based model for continental weathering to be coupled with the present model will lead to a better understanding of controls on oceanic $\delta^{18}$O during the Precambrian.

## 4.3 Comparison with other models

Our model results contrast with the results by Lécuyer and Allemand (1999), who utilized prescribed distributions of temperature and water/rock ratio and an isotope-fractionation equation for a half-closed system (a system that is open for water phase,





but not for solid phase; Gregory et al., 1989) (Introduction). The assumed range for water/rock ratio ($\leq 10$) by these authors is comparable to the simulated range in the present study ($\eta \leq 164$; Figs. 5a and c). Also, because the Arrhenius equation adopted

for kinetics of oxygen isotope exchange by Lécuyer and Allemand (1999) is based on the dataset by Cole and Ohmoto (1986), the kinetic expression in the present study (Eq. (7); based on Cole et al., 1983, 1987) should not be significantly different from that by Lécuyer and Allemand (1999). States of equilibrium and non-equilibrium in the bulk-rock-based isotope exchange formulation in the present study correspond to transport- and reaction-limited alteration states, respectively, in the mineral-based isotope exchange formulation in Lécuyer and Allemand (1999); this formulation difference should not cause any significant

differences in oxygen isotope behavior for bulk rock and porewater. Additional simulations with a wider calculation domain 300 km (cf. 30 km in the standard simulation) and artificially imposing off-axis water exchange (up to $5.3 \times 10^{15}$ kg yr$^{-1}$) do not show significant differences from the results presented above (Supplementary material); the size difference in the calculation domain cannot explain the difference between the present study and Lécuyer and Allemand (1999). Although a thorough comparison cannot be made because Lécuyer and Allemand (1999) have not provided detailed model results to be compared

with those presented here, it is likely that the isotopic effect of spreading crust has not been explicitly considered by Lécuyer and Allemand (1999). Should this be the case, the sensitivity of oceanic rocks to seawater $\delta^{18}$O and the buffering capacity at midocean ridges could be overestimated in Lécuyer and Allemand (1999). This overestimation could also be applicable to other models, e.g., by Taylor (1977), Norton and Taylor (1979), Criss et al. (1987), Gregory et al. (1989) and DePaolo (2006), who did not explicitly consider the transport of solid rocks, either.

**5 Conclusions**

The 2D reactive transport model of oxygen isotopes combined with 2D hydrothermal circulation simulations enables us to predict distributions of temperature, water flow and oxygen isotopes of solid rocks and porewaters within oceanic crust based on mass, momentum and energy conservations. The model assuming the present-day seawater $\delta^{18}$O reproduced those distributions consistent with modern observations, supporting the model's validity. The intensity of seawater-$\delta^{18}$O buffering at midocean

ridges was evaluated by calculating the net $^{18}$O flux as a function of seawater $\delta^{18}$O. The buffering intensity predicted by the model is significantly weaker than those previously assumed under any plausible alteration conditions during the Precambrian. The weak buffering is realized because isotope exchange equilibrium is not reached in shallow sections of oceanic crust with low temperatures, and because $^{18}$O supply via spreading solid rocks exceeds that through hydrothermal circulation in deeper high-temperature sections. Consistently with the weak buffering, oceanic rocks are insensitive to seawater with respect to

oxygen isotopes. Thus, ophiolites can alternatively be explained by the insensitivity of oceanic rocks to seawater $\delta^{18}$O that could have evolved through the Precambrian because of the weak buffering at midocean ridges.

*Code availability.* The source codes of the model are available upon request to the author.





## Appendix A: Supplementary material

Supplementary material related to this article can be found online at https://doi.org/xxxxxx.

*Author contributions.* YK conduced the whole research.

*Competing interests.* I declare no competing interests.

*Acknowledgements.* I thank Jim Kasting and Lee Kump for helpful discussions. Laurence Coogan and an anonymous reviewer of earlier version of manuscript provided helpful comments.





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



Table 1. Symbols and their definitions and values.

| Symbol | Definition | Value |
|---|---|---|
| $c_{\mathrm{p}}^{\mathrm{f}}$ | Specific heat capacity of water (J kg$^{-1}$ K$^{-1}$) | |
| $c_{\mathrm{p}}^{\mathrm{m}}$ | Specific heat capacity of oceanic rock (J kg$^{-1}$ K$^{-1}$) | $10^3$ |
| $D$ | Effective diffusion coefficient for $^{18}$O (m$^2$ yr$^{-1}$) | |
| $E$ | Apparent activation energy of oxygen isotope exchange (J mol$^{-1}$) | $5 \times 10^4$ |
| $F_{\mathrm{m}}$ | Mole ratio of $^{18}$O to total O ($^{18}$O plus $^{16}$O) of pristine crust (dimensionless) | $2.0126 \times 10^{-3}$ |
| $F_{\mathrm{p}}$ | Mole ratio of $^{18}$O to total O ($^{18}$O plus $^{16}$O) of porewater (dimensionless) | |
| $F_{\mathrm{r}}$ | Mole ratio of $^{18}$O to total O ($^{18}$O plus $^{16}$O) of solid rock (dimensionless) | |
| $F_{\mathrm{sw}}$ | Mole ratio of $^{18}$O to total O ($^{18}$O plus $^{16}$O) of seawater (dimensionless) | |
| $g$ | Gravity acceleration (m yr$^{-2}$) | $9.76 \times 10^{15}$ |
| $\mathbf{g}$ | Gravity vector (m yr$^{-2}$) | |
| $J_{\mathrm{b}}$ | Boundary heat flux from the intrusion (J m$^{-2}$ yr$^{-1}$) | |
| $k$ | Permeability of oceanic rock (m$^2$) | |
| $k_{\mathrm{ex}}$ | Rate constant for oxygen isotope exchange between solid rock and porewater (mol$^{-1}$ kg yr$^{-1}$) | |
| $m_{\mathrm{f}}$ | Mole concentration of oxygen per unit water mass (mol kg$^{-1}$) | 55.56 |
| $m_{\mathrm{s}}$ | Mole concentration of oxygen per unit solid mass (mol kg$^{-1}$) | 31.25 |
| $P$ | Fluid pressure (Pa) | |
| $\mathbf{q}$ | Water mass flux vector (kg m$^{-2}$ yr$^{-1}$) | |
| $R_{\mathrm{g}}$ | Gas constant (J mol$^{-1}$ K$^{-1}$) | 8.314 |
| $t$ | Time (yr) | |
| $T$ | Temperature (K) | |
| $T_{\mathrm{m}}$ | Temperature of the intrusion (K) | $1.473 \times 10^3$ |
| $w$ | Spreading rate of oceanic crust (m yr$^{-1}$) | |
| $x$ | Distance from ridge axis (m) | |
| $y$ | Depth from ocean/crust interface (m) | |
| $\alpha$ | Oxygen isotope fractionation factor (dimensionless) | |
| $\beta$ | Parameter relevant to the temperature dependence of $\alpha$ (dimensionless) | 0.876 |
| $\eta$ | Water/rock oxygen-mole ratio (dimensionless) | |
| $\kappa$ | Thermal conductivity of oceanic rock (J yr$^{-1}$ m$^{-1}$ K$^{-1}$) | $9.47 \times 10^7$ |
| $\mu$ | Kinematic viscosity of water (m$^2$ yr$^{-1}$) | |
| $\rho_{\mathrm{b}}$ | Density of bulk rock (kg m$^{-3}$) | |
| $\rho_{\mathrm{f}}$ | Density of water (kg m$^{-3}$) | |
| $\rho_{\mathrm{m}}$ | Density of solid rock (kg m$^{-3}$) | $3 \times 10^3$ |
| $\tau$ | Tortuosity factor (dimensionless) | |
| $\phi$ | Porosity of crust (dimensionless) | $5 \times 10^{-2}$ |
| $\Omega$ | Degree of oxygen isotope exchange (dimensionless) | |
| $\nabla$ | Vector differential operator (m$^{-1}$) | |



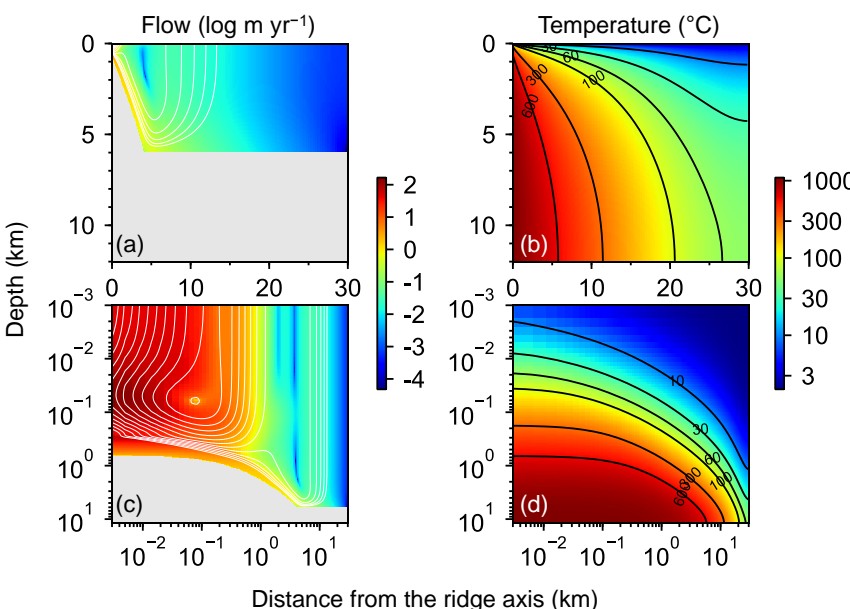

Figure 1. Two-dimensional distributions of hydrothermal fluid flow (a and c) and temperature (b and d). Shown in a and c are logarithms of fluid velocity $(\mathrm{m\ yr^{-1}})$ together with mass-based stream lines that are depicted with white curves. The same data as in a and b are ploted on logarithmic scales in c and d, respectively. Gray zones in a and c represent where rocks are impermeable below 6 km depth and/or with temperatures above the rock-cracking threshold (600 °C).



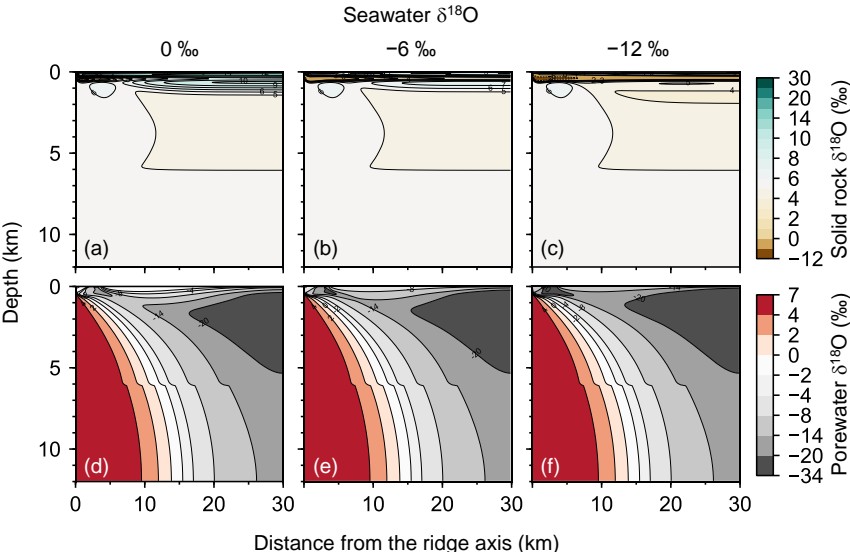

Figure 2. Two-dimensional distributions of solid-rock and porewater $\delta^{18}$O (a–c and d–f, respectively) at 0, −6 and −12 ‰ of seawater $\delta^{18}$O (a and d, b and e, and c and f, respectively).





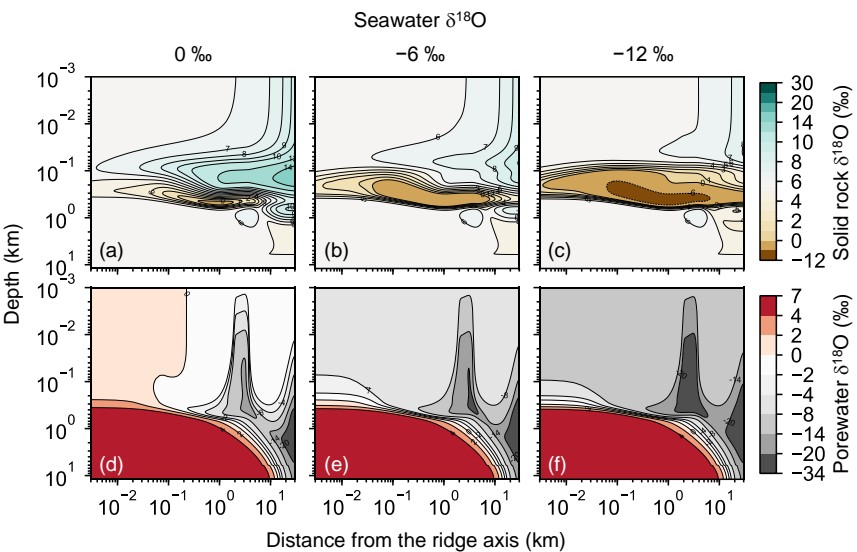

Figure 3. As for Fig. 2, except plotted on logarithmic scales.

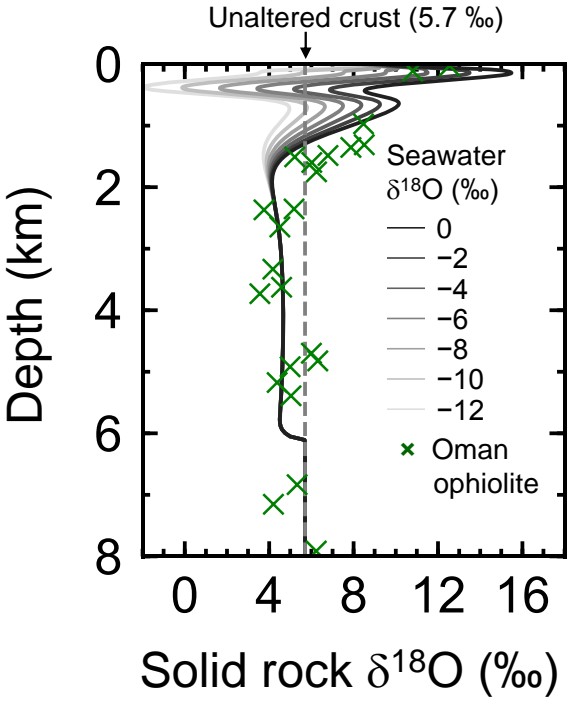

Figure 4. Solid rock $\delta^{18}$O as function of depth at 1 Ma (30 km) from the ridge axis with $0, -2, ..., -12$ ‰ of seawater $\delta^{18}$O. Also plotted are Oman ophiolite data from Gregory and Taylor (1981) (crosses) converting their reported depth $y'$ (km) to $(7.4 - y')$ (km) to facilitate comparison.

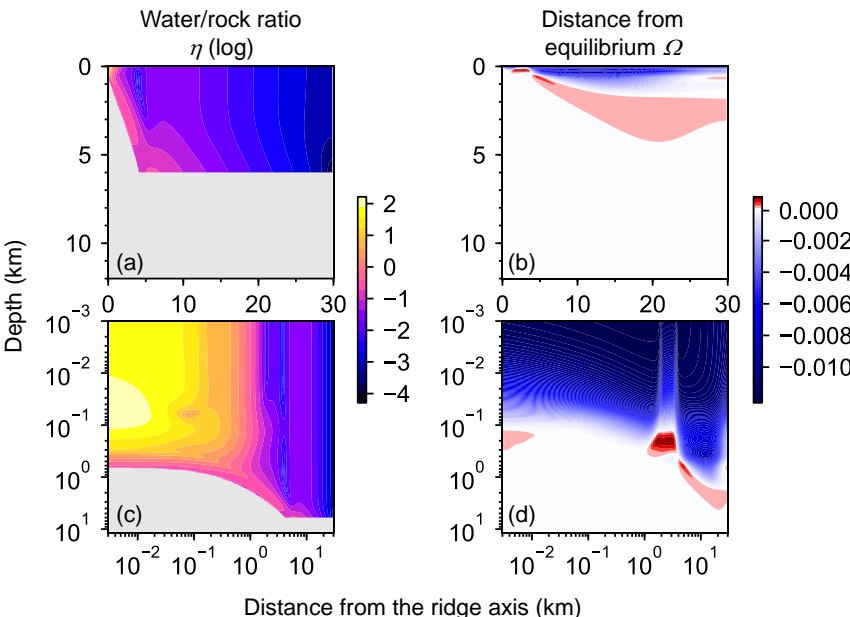

Figure 5. Two-dimensional distributions of local water/rock oxygen-mole ratio (a and c) and degree of oxygen isotope exchange at 0 ‰ of seawater $\delta^{18}$O (b and d). The same data as in a and b are ploted on logarithmic scales in c and d, respectively. See the caption of Fig. 1 for the explanation of gray zones in a and c.





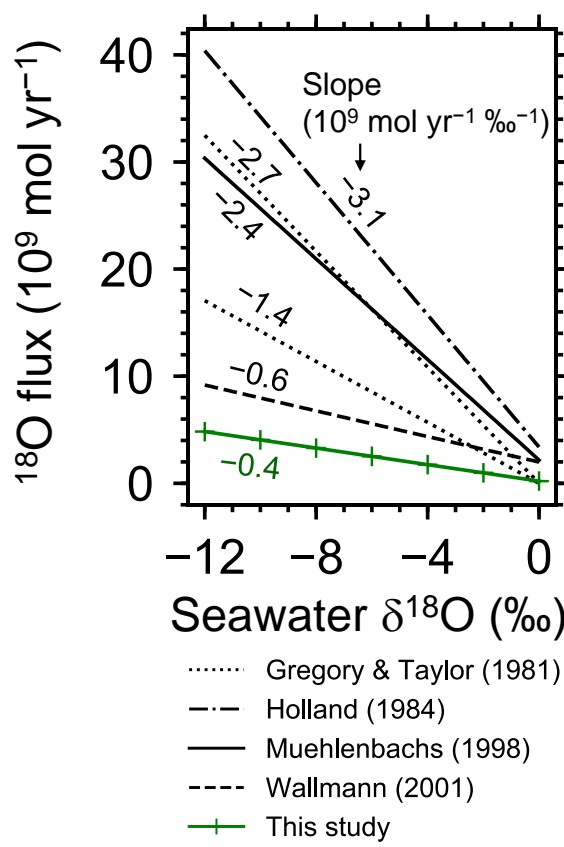

Figure 6. Net $^{18}$O flux to the ocean from hydrothermal systems as function of seawater $\delta^{18}$O. Slope values are denoted near the lines from previous studies and the standard simulation in this study.



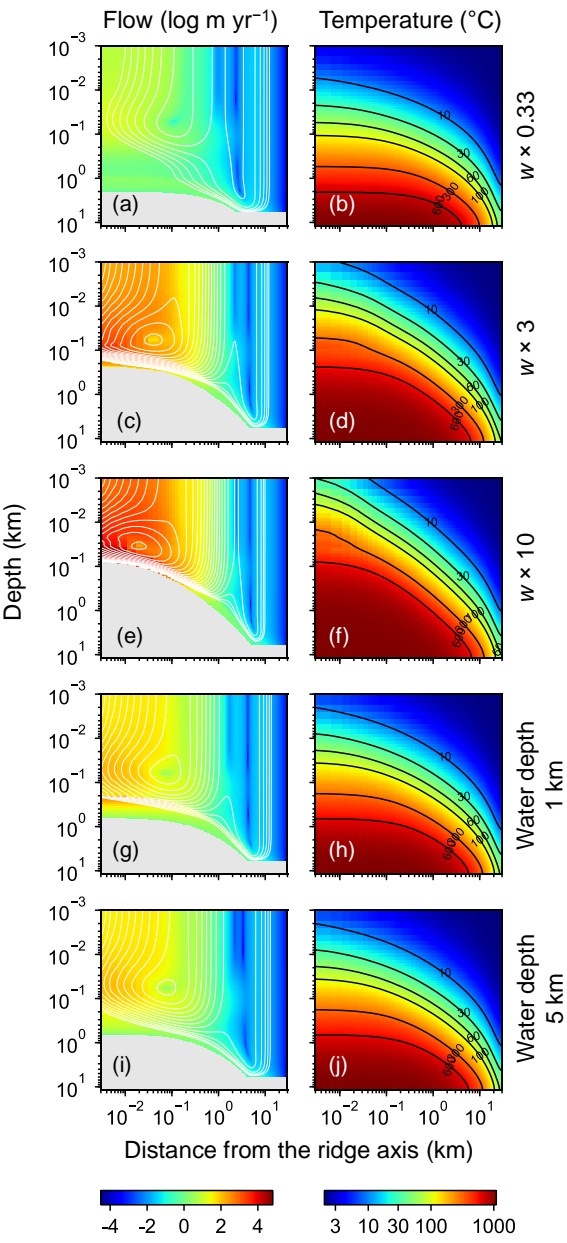

Figure 7. Two-dimensional distributions of hydrothermal fluid flow (a, c, e, g and i) and temperature (b, d, f, h and j) from simulations with different spreading rates and ocean depths. Logarithms of fluid velocity and mass-based stream lines are depicted in a, c, e, g and i. Spreading rate is changed at $1 \times 10^{-2}$ m yr$^{-1}$ in a and b, $9 \times 10^{-2}$ m yr$^{-1}$ in c and d, and $30 \times 10^{-2}$ m yr$^{-1}$ in e and f, while ocean depth is changed at 1 km in g and h and 5 km in i and j. Values of other parameters are the same as those in the standard simulation (e.g., Fig. 1). See the caption of Fig. 1 for the explanation of gray zones in a, c, e, g and i.

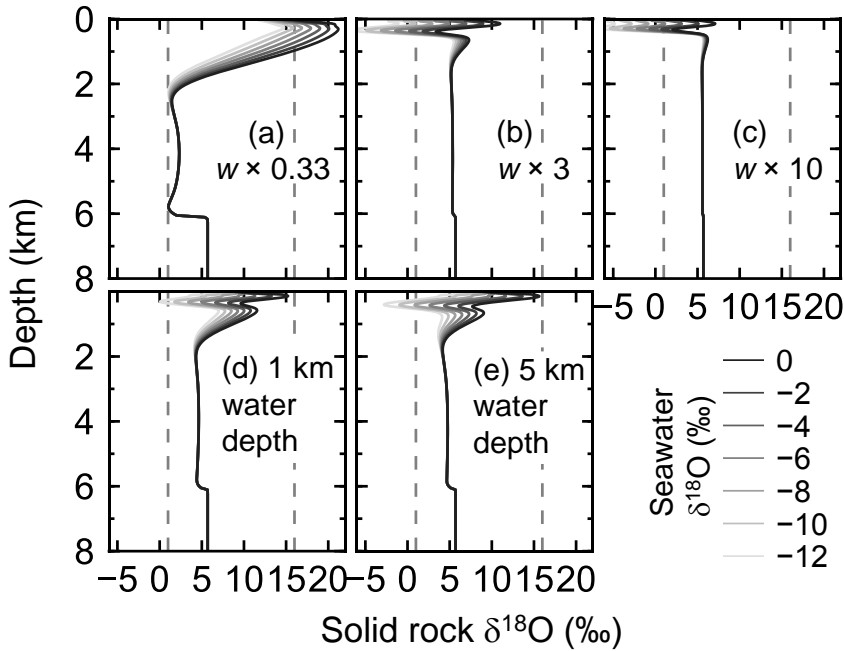

Figure 8. Solid rock $\delta^{18}O$ as function of depth at 30 km from the ridge axis with $0, -2, ..., -12\,‰$ of seawater $\delta^{18}O$ from simulations with different spreading rates and ocean depths. Dashed lines denote 1 and 16 ‰, between which observed $\delta^{18}O$ of ophiolites and/or oceanic crust ranges (Gregory and Taylor, 1981; Barrett and Friedrichsen, 1982; Cocker et al., 1982; Elthon et al., 1984; Alt et al., 1986, 1995; Agrinier et al., 1988; Schiffman and Smith, 1988; Vibetti et al., 1989; Lécuyer and Fourcade, 1991; Stakes, 1991; Bickle and Teagle, 1992; Holmden and Muehlenbachs, 1993; Muehlenbachs et al., 2003; Alt and Bach, 2006; Furnes et al., 2007; Gao et al., 2012). Standard parameter values are assumed except for the parameterization denoted at each panel.



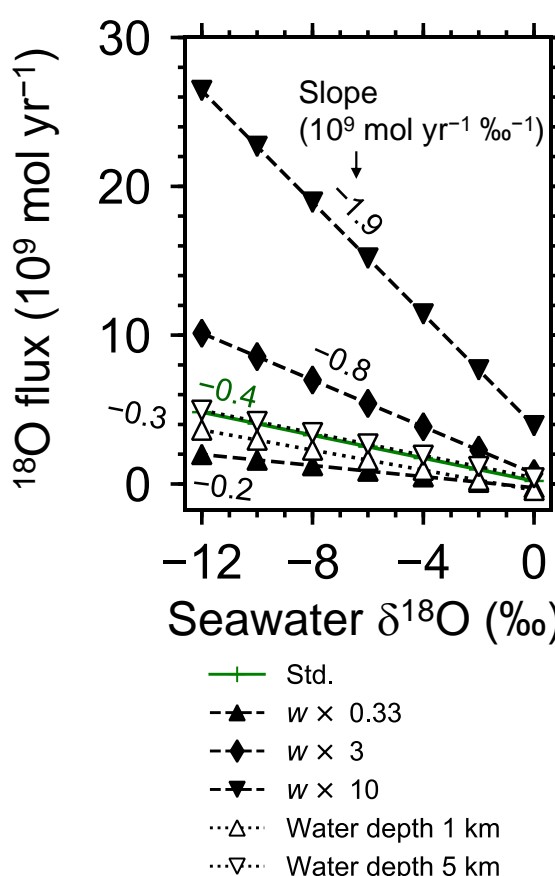

Figure 9. Net $^{18}$O flux to the ocean from hydrothermal systems as function of seawater $\delta^{18}$O from simulations with different spreading rates and ocean depths. See the legend for the types of symbol and line for individual simulations. Slope values are denoted near the lines.