# Peer review of "Quantifying the buffering of oceanic oxygen isotopes at ancient midocean ridges"

_Solid Earth, 2019_

## Referee Comment (RC1) · Benjamin Johnson (Referee) · 14 Mar 2020

Review of Quantifying the buffering of oceanic oxygen isotopes at ancient mid-ocean ridges:
Yoshiki Kanzaki

*General Comments.* The author uses a coupled hydrothermal alteration-reactive transport
model to investigate how oxygen isotopes are buffered during alteration of ocean crust. There
was been a renewed interest in the controls on and history of the oxygen isotope composition
of seawater, and this is a timely manuscript. The main conclusion from the work is that the
buffering capacity is lower than previously expected, primarily due to slow kinetics of isotope
exchange at low temperatures and shallow crustal depths. This was tested over a range of
seawater $\delta^{18}O$ values, and the work indicates that patterns of alteration in ocean crust are
similar across this range. The author suggests, then, that the apparent constancy of ophiolite
$\delta^{18}O$ through time does not, in fact, require constant seawater $\delta^{18}O$ through time. Rather, this
crustal record is not a good reflection of seawater $\delta^{18}O$.

Thank you for the opportunity to read this paper! I think it's a valuable contribution, but I do
have some comments below. Primarily, I'm interested in how you setup your model and some
implications:

- What is the importance of pore-water exchange vs fluid in cracks? My impression was
  that more water is transported through cracks than pores?
- Would you expect this relationship if seawater had a positive $\delta^{18}O$, as has been
  suggested in previous and recent work? (Johnson and Wing, 2020, *Nature Geoscience,*
  Pope et al., 2012, *PNAS*).
- In your Figure 4, it looks like the measurements from Oman most closely match your
  simulation from a 0‰ ocean. The upper part of the crust, from your model, does
  change quite  a bit under different ocean  $\delta^{18}O$. There are older ophiolites that you
  could compare here, such as the one from Holmden and Muehlenbachs (1993), or
  Muehlenbachs et al. (2003). This figure makes it seem like the upper part of the crust is
  in fact sensitive to changing seawater  $\delta^{18}O$, so couldn't it actually be used as a proxy for
  seawater $\delta^{18}O$?

*Specific Comments.*
- Paragraph lines 26-42: In addition, lower temperatures are supported by O-isotopes in
  phosphates (Blake et al., 2010 Phosphate oxygen isotopic evidence for a temperate and
  biologically active Archaean ocean), so it's not just sporadic glacial activity. There are
  also GCM studies that support non-super hot conditions (Wolf and Toon, 2014,
  Controls on the Archean Climate System investigated with a global climate model)
- In this same paragraph, it's important to note that the samples from the new Galili et
  al. study are all from the Proterozoic and younger, and do not give additional
  information on the Archean.
- Line 105-106: is $10^4$ years sufficient? Many low-temperature systems last much longer
  than this, with additional water circulation
- Equation 7: This seems to be a key part of your conclusions, that slow kinetics limit O-
  isotope buffering. Your constant, $10^{-8.5}$, is lower than previous estimates. This value
  needs a bit more justification. What is the reasoning that field kinetics are slower? Is it
  just harder to measure?
- In addition, the related material in the supplement (Fig. S7), appears to show a pretty
  different pattern of  $\delta^{18}O$ in the crust depending on $k_{ex}$. Can you provide some
  additional justification?

- I grant that your model fits the Oman ophiolite data well, but we know that the $\delta^{18}O$ of seawater at the time this formed is not different than today, so perhaps testing your model in a system that we **know** has a different $\delta^{18}O$ value, such as a freshwater system, might be insightful.
- Equation 8: why use this equation for andesite? You say it's similar to Cole et al. for basalt, so what is the advantage?
- Why is permeability set to 0 below 6km?
- Depaolo (2006), which you do cite, found that equilibrium exchange is a good approximation as long as fractures are ~1-4 m apart, as in MOR. Why does your work differ here?
- Another study using a similar approach is Cathles, L. M. in *The Kuroko and Related Volcanogenic Massive Sulfide Deposits* Vol. 5 (eds Ohmoto, H. & Skinner, B. J.) 439–487 (Economic Geology Publishing, 1983). How does your work compare to theirs, which is very similar in approach?

*Technical Comments.*
- Typo in line 33? Should this be 70-85 degrees C?

---

## Referee Comment (RC2) · Itay Halevy (Referee) · 16 Apr 2020

Review of: Quantifying the buffering of oceanic oxygen isotopes at ancient midocean ridges By: Yoshiki Kanzaki

Recommendation: Major revision.

**Manuscript summary:**

The author develops a 2D coupled model of hydrothermal circulation and oxygen isotope exchange in oceanic crust, and uses the model to address the apparent disagreement between d18O records in authigenic marine precipitates and altered oceanic crust sequences ("ophiolites"). The author validates the model against several observations and estimates of heat fluxes, water fluxes, d18O profiles in an ophiolite, etc. The author then uses the model to investigate the capacity of hydrothermal circulation and alteration of the oceanic crust to buffer the isotopic composition of seawater. The main finding is that the d18O profiles in the altered crust and the resulting net O isotope fluxes to/from the ocean are insensitive to seawater d18O, relative to suggestions in previous studies, suggesting that seawater d18O is very weakly buffered by hydrothermal alteration of oceanic crust. This weak feedback is suggested to arise from a combination of kinetically limited O isotope exchange in the cooler portion of the crust, and rock-dominated O isotope exchange in the deeper, hotter portion of the crust. The author explores the sensitivity of the feedback strength to several physical parameters and model design choices, and finds that the weak feedback is robust to these choices. The conclusion is that ophiolite d18O profiles that are invariant over Earth history cannot be used to infer constant seawater d18O through time.

**Review summary:**

This is a well-written and reasoned manuscript, which addresses an important and actively debated topic: the evolution of seawater O isotope ratios over Earth history. The approach of coupling 2D forward models of heat and water transport and of O isotope exchange is innovative, and it promises to bring constraints and insights of a different nature to this debate. I have only two major comments, which appear in several of my specific comments below.

Firstly, where the model is validated against ophiolite d18O profiles or estimates of 18O fluxes to/from the oceanic crust, these consistency tests only have meaning when the age (i.e., alteration duration) of the oceanic crust is known, and when the model results at that specific alteration duration are compared to the observations. The model needs to be validated against profiles with better-constrained duration of alteration, perhaps from ODP boreholes.

Secondly, all of the insights gained from the model are based on simulation of circulation and O isotope exchange out to a distance of 30 km from the ridge axis, and a more limited investigation of off-axis alteration out to 300 km. The claims made on the basis of these simulations have far-reaching implications. In my opinion, an effort should

be made to show that the weak buffering intensity revealed by the model is not an outcome of this limited model spatial domain. In other words, if one considers sustained low-T alteration as the crust continues aging and until it is subducted, do the main findings of this study hold? Are O isotope fluxes still insensitive to seawater d18O? I urge the author to test this, which will provide confidence in the findings.

Finally, not a concern so much as a suggestion, related to my second major comment. If this detailed modeling reveals an insensitivity to seawater d18O even out to thousands of km from the ridge axis, but a dependence of subducted crust d18O on physical parameters such as the spreading rate and the thickness of sediment draped on the oceanic crust, then it may provide an explanation not only for the invariant d18O of ophiolites, but also for the long-term secular evolution of seawater d18O. Perhaps this is beyond the scope of the current study, but it would be a welcome and timely contribution.

On the basis of the volume of work required to validate the model against wellconstrained targets and to model hydrothermal alteration out to larger distances from the ridge axis, I recommend major revision. Once revised, this study will be an important contribution to the outstanding debate about seawater O isotopes.

Below are my specific comments. Comments that are related to my main comments are in **bold**.

Itay Halevy

**Specific comments:**

1. L19-21: The sentence in these lines can be worded more clearly.

2. L26: It may be worth mentioning that by "authigenic sedimentary rocks" you mean d18O records in carbonate rocks, cherts, phosphorites, glauconites and shales, all of which show a pronounced increase in d18O over Earth history.

3. L32: Is the range 70–15°C correct? Shouldn't the second number be larger than 70?

4. L38: Perhaps "weak" instead of "little"?

5. L38-42: The motivation for reconciling the sedimentary and ophiolite records is more than just being able to use sedimentary d18O records to reconstruct temperatures. The evolution of seawater d18O is driven by the same processes that govern the chemical fluxes to the ocean (e.g., from low-T continental weathering, from hydrothermal alteration of the oceanic crust at both high and low T), with implications for the evolution of ocean chemistry, the attendant productivity of the biosphere and the composition of the atmosphere.

6. L52-54: The statement in this sentence is not entirely correct. Any mechanism to lower the T of oceanic crust alteration will result in greater enrichment of the altered crust in 18O (and greater removal of this 18O from the ocean, as the author mentions).

For a given amount of alteration, the resulting altered crust will be more strongly offset in d18O from the altering fluid (~seawater). In this case, more 18O-depleted seawater gives rise to correspondingly 18O-depleted authigenic minerals, as observed. This same 18O-depleted seawater could concurrently give rise to altered oceanic crust with d18O similar to modern altered oceanic crust, if the alteration T was lower and the mineral-water O isotope fractionation larger. So at least some of these mechanisms may also explain approximately invariant ophiolite d18O.

7. L67: "The present study has been undertaken to present...". Suggest rewording.

8. L95: Should be "length scale"? Also, perhaps "for an e-fold increase"?

9. L107: Why does the model grid extend to 12 km if the rocks are taken to be impermeable below 6 km?

10. L107: The choice of a domain length of 30 km from the ridge axis has implications for the timescale of the simulation. At the spreading rates investigated in this study (1e-2 to 30e-2 m/y), formation of 30 km (3e4 m) of new crust takes between 1e5 and 3e6 years. How long does it take to reach a steady state for the q-*P*-*T* fields over the domain? Presumably much less than the time that it takes newly produced crust to exit the model domain?

11: L112-113: When you refer to the bottom and right boundaries as insulating, do you mean that there is no temperature gradient across these boundaries? Do the results change if you relax this assumption (e.g., using Neumann boundary conditions with a non-zero flux)? You mention what happens when you relax the assumption of impermeability of these boundaries, and it would be good to also mention what happens when you don't assume the boundaries to be insulating.

12. L140-143: Could you please better substantiate the choice of a lower-than-lab kex? Is it only due to the smaller specific surface area in the field, or are there other factors, too? In the Supplementary Material, it would be good to show the sensitivity to kex up to the highest lab values (10^-6.6 mol/kg y). This would increase confidence in the low d18O buffering capacity of seafloor alteration suggested in this study.

13. L151-152: "The first term on the right-hand side..., while the second term represents the hydrodynamic dispersion.".

14. L158: Is the O isotope model insensitive to the assumption of impermeability, like the  $\mathbf{q}$ -*P*-*T* model?

15. L163: The Results section contains a lot of discussion. It may be useful to combine the Results and Discussion sections.

16. L168 and elsewhere: "Ma" is usually reserved for millions of years ago. When referring to millions of years, "Myr" is more commonly used.

17. L170: The modeled water mass flux is not only within the range of Elderfield and Schultz (1996), it is quite close to their recommended value of  $3(\pm 1.5)$  e13 kg H2O/y.

18. L172-174: There is nothing special about the distance of 30 km from the midocean ridge - if alteration is a sustained process, then there will be some distance at which the model d180 profiles most closely resemble the observations. For a different spreading rate, "consistency with observations" could be reached at a different distance from the spreading center, as suggested by Fig. 8. Consistency can be assessed (or the model calibrated, alternatively) only with independent knowledge on the age of the profiled crust - how long was the sampled crust altered, and does the model resemble the d180 profile in that crust at a comparable duration of alteration. The model should be tested against d180 profiles in crust with a known duration of alteration (perhaps in ODP boreholes).

The above relates to a bigger issue, which is the somewhat arbitrary choice of 30 km as the edge of the model domain. Does alteration of the oceanic crust stop farther out from the spreading center? Again, Fig. 8 suggests that this is not the case. In panel (a) of that figure a lower spreading rate results in much more 180-enriched altered crust than at higher spreading rates (Fig. 4, 8b, 8c). Would this degree of enrichment not be reached farther out from the spreading center at the higher spreading rates? Does the proposed insensitivity to seawater d180 hold if alteration continues over the lifetime of an oceanic plate?

To address this, the author should perform simulations out to much greater distances from the spreading center and identify the distance from the ridge at which the isotopic composition no longer changes. I presume this distance will depend on the model parameters, and this may affect the sensitivity of the ultimate isotopic composition of the crust on seawater d180. I don't know if this request is practical, given the computational cost of extending the simulation out to thousands of km from the ridge. If not, a way to parameterize the behavior farther away from the ridge with continued water-rock interaction and O isotope exchange should be developed.

As an aside, constraints on the distance to which water-rock interactions continue to change the isotopic composition of oceanic crust have implications for the effect of oceanic crust alteration on the isotopic composition of seawater. For example, if alteration continues over much of the lifetime of an oceanic plate, then slower seafloor spreading in the Precambrian, as suggested in several recent studies (several papers from Korenaga over the past decade; Fuentes et al., 2019), would lead to subduction of older, more 180-enriched crust, leaving the ocean 180-depleted (Galili et al., 2019).

19. L178-179: The sentence in these lines is difficult to understand. Suggest rephrasing.

20. L180-189: The model d18O profiles in the bulk rock and the 18O fluxes from high- and low-T alteration are reported in these lines and compared with available

observations and previous estimates. As in comment #18, consistency with the observed profiles has meaning only if the model and observed profiles are of an equivalent age (i.e., alteration duration). Likewise, the consistency between model 180 fluxes and previous estimates has meaning only if the estimates were made on the basis of altered crust of a comparable age.

21. Sections 3.2, 3.3 and onwards: The results, interpretations and implications in the rest of the manuscript should be consistent with the tests performed in response to comments #18 and 20 above.

**22. L207-212: As in comments #18 and 20, does the distance from equilibrium keep decreasing past 30 km? If it keeps decreasing, does this affect the proposed insensitivity to seawater d180?**

23. L213: Perhaps it would be useful to mention that the reason for the near-equilibrium in the deeper parts of the section are due to the higher T.

24. L238: Perhaps change "not inconsistent with" to "consistent with"?

25. L235-238: The way these results are reported is very hard to take in, with all of the numbers and parentheses within parentheses. Suggest rewording.

26. L252: "spreading" and "weaker" are misspelled.

27. L255-278: The two paragraphs in these lines are less well-written than the previous text. Suggest editing for grammar, language and clarity.

28. L273-278: The model of Kasting et al. (2006) included the effect of overburden (ocean depth) on the depth in the crust at which water reached the critical point, leading to changes in the capacity of hydrothermal systems to transport heat and, consequently, on the temperature profile of water-rock interactions. Are such water phase changes considered in the present model, and if not, could that be an additional reason for disagreement with the results of Kasting et al. (2006)? Please discuss.

**29. L281-282: This statement needs to be reevaluated following the tests requested in comments #18, 20, 21, 22. Hopefully, it still holds.**

30. L283-286: This sentence is awkwardly worded. Suggest "By comparison, the simulated solid rock d18O values fall within this range for seawater d18O values  $\geq$ -10, -8 and -2‰ at a spreading rate of 1e-2, 3e-2 and  $\geq$ 9e-2 m yr^-1, respectively (Figs. 4, 8)." Related to the above, what are the average Archean/Proterozoic/Phanerozoic seafloor spreading rates suggested in previous studies, and what are the implications for the evolution of the 18O-buffering strength of hydrothermal alteration of oceanic crust over Earth history?

31. L286-288: There are values of seawater d18O that are inconsistent with the range observed in ophiolites, right? Perhaps mention those values? Related to this, it appears

that the model reproduces the range observed in ophiolites irrespective of seawater d18O mostly at low spreading rates. It is worth mentioning that estimated Precambrian seafloor spreading rates were slower than Phanerozoic rates.

32. Section 4.3: This section could also benefit from editing for grammar, language and clarity.

33. Fig. 1: The labels on contours in panels b and d can be moved and spread out so that they are more easily seen. In panel b, orienting the text sub-parallel to the contours near the bottom and right domain boundaries would work nicely. In panel d, orienting the text sub-parallel to the contours near the left boundary would work.

34. Maybe it's just on my laptop, but there are fine horizontal and vertical lines on the filled contour plots with a continuous color scale (Fig. 1, 5, 7).

35. Fig. 2: Suggest changing "0, –6 and –12  $\infty$  of seawater d18O" to "at seawater d18O values of 0, –6 and –12  $\infty$ ".

36. Fig. 4 caption: "Ma" -> "Myr". Suggest changing "0, -2, ..., -12 ‰ of seawater d18O" to "at seawater d18O values of 0, -2, ..., -12 ‰". Note that this comparison is meaningful only for crust of a similar alteration duration (see comments #18, 20, 21, 22).

37. Fig. 5 caption: "0 ‰ of seawater d18O" -> "a seawater d18O value of 0 ‰".

38. SM L33: As mentioned in comment #12, the choice of a factor of 10 for the uncertainty is arbitrary. It would be good to perform an additional simulation at kex =  $10^{-6.5}$ . If the results are indeed insensitive to the value of kex, this will not matter much for the buffering intensity, and it would provide confidence in the proposed insensitivity of seafloor alteration to seawater d18O.

39. Fig. 7 caption: The sentence starting with "Spreading rate" is awkward. Suggest rewording.

40. Fig. 8: Suggest decreasing font size of axis tick labels. Also, "0, -2, ..., -12 ‰ of seawater d18O" -> "at seawater d18O values of 0, -2, ..., -12 ‰".

41. SM Section S3, Figs. S7, S8: Looking at Fig. S7, there are significant differences between the profiles at a different value of kex. Please explain mechanistically why the buffering intensity ends up being so similar.

42. SM Section S4: A major concern of any clued reader will be that the current model only extends out to an oceanic crust age of 1e5 to 3e6 years (see many of my comments above). As such, I suggest moving some of this section to the main text, perhaps in the discussion.

43. SM Section S4: Is a distance of 300 km from the ridge axis sufficient? Does the model d180 of the crust stop evolving after this distance? As with many of

my comments above, it is important to constrain the change in the profiles as the crust ages and run the simulations out to a distance beyond which the additional change is negligible.

44. SM Section S4: The finding that the buffering intensity is no different from the standard case when off-axis alteration is included is very important, and it is understandable that the author focuses on this aspect, given the focus of the paper. However, there is a missed opportunity here, in my opinion, which is an exploration of ways in which changes through Earth history in seafloor spreading rates and oceanic plate lifetimes affect the net budget of 180. Fig. S13 clearly shows that despite similar buffering intensities, the cases with off-axis circulation differ substantially in the net 180 flux from the standard case. If the proportion of off-axis alteration out of the total alteration has changed through time (e.g., changing spreading rate, changing sediment cover, changing crustal thickness), the current model can help to explain the change in seawater d180 suggested on the basis of the 0 isotope record in authigenic minerals. Perhaps this is beyond the scope of the current contribution.

45. SM L44-46: Please elaborate on the basis for the notion that the oceanic crust is altered within 10 Myr of its formation. The author's off-axis simulations suggest continued low-T alteration for much longer durations.

46. SM L51: What is the approximate sediment thickness required for this additional 10 MPa? With a density of 2700 kg/m^3 and an assumed porosity of 0.5, about 550 m of sediment are required. Please comment on the plausibility of this at 300 km from the spreading center (given, e.g., Straume et al., 2019) - to me this seems high. Fisher and Becker applied pressures  $\leq$ 1-3 MPa, up to an order of magnitude less than here. Is it possible to overcome the numerical issues and perform the off-axis simulations with less of an overburden and lower imposed pressures?

47. SM Section 4 and elsewhere: Please replace "Ma" with "Myr", as necessary (see comment #16).

48. SM L74-78: See comment #44. There is a missed opportunity here.

49. SM Fig. S1 caption: "0, -6 and -12 % of seawater d18O" is grammatically awkward. I suggest changing this (in two places in the caption) to "at seawater d18O values of 0, -6 and -12 %". Likewise, suggest "adopt a spreading rate of R1, R2 and R3, respectively." instead of the current text.

50. SM Fig. S2 caption: Same as comment #49. This wording appears also in several of the other SM figures. Suggest changing.

51. SM Fig. S3, S5, S6, S7, S8, S12, S13: Suggest smaller font size on axis tick labels.

---

## Author Comment (AC1) · 10 Jun 2020

**Response to Referee #1 (Dr. Benjamin Johnson)**

I express my gratitude to Dr. Benjamin Johnson for his useful comments. My response to the reviewer's comments and the corresponding revision are described in detail below. The numbers of pages, lines, equations, tables and figures are those in the revised manuscript unless otherwise described.

General comment 1:
"What is the importance of pore-water exchange vs fluid in cracks? My impression was that more water is transported through cracks than pores?"

Response:
As long as fractures/cracks occur on smaller spatial scales than the control volume in the calculation domain, their effects can be accounted for by adopting corresponding bulk-rock permeability (cf. Cathles, 1983; DePaolo, 2006). DePaolo (2006) suggested 1-4 m for the fracture spacing, which is generally smaller than the grid cells of the calculation domain ($>\sim 1$ m in horizontal). Therefore, it is not unreasonable to account for the presence of factures/cracks by adopting correspondingly high permeability for the bulk rock. The permeability at the crust/ocean interface is assumed to be $\sim 10^{-12}$ $m^2$, which falls in the range of fractured rock permeability ($\geq 10^{-12}$ $m^2$; Fisher, 1998). Also, the model that assumes a higher permeability ($\sim 10^{-11}$ $m^2$ at the ocean/crust interface) yields essentially the same results as those in the standard simulation (Supplementary material). Therefore, the present results and conclusions will remain valid in systems that include fractures/cracks.

Changes in manuscript (Page numbers/Line numbers):
Description of fractured rock permeability is added to Section S3 in Supplementary material where I compare the permeability adopted for this study and observations by Fisher (1998) (P3/L72-73 in Supplementary material).

General comment 2:
"Would you expect this relationship if seawater had a positive δ18O, as has been suggested in previous and recent work? (Johnson and Wing, 2020, Nature Geoscience, Pope et al., 2012, PNAS)."

Response:
I ran additional experiments that assume positive seawater $\delta^{18}O$ up to 6 ‰ and confirmed that the results and conclusions in the manuscript remain valid.

Changes in manuscript (Page numbers/Line numbers):

I modified figures to include the results from simulations that assume positive seawater $\delta^{18}O$ (P23, P25, P27, P28) and included the reference of Johnson and Wing (2020) in the revised manuscript (P9/L252-253).

General comment 3:

"In your Figure 4, it looks like the measurements from Oman most closely match your simulation from a 0‰ ocean. The upper part of the crust, from your model, does change quite a bit under different ocean δ18O. There are older ophiolites that you could compare here, such as the one from Holmden and Muehlenbachs (1993), or Muehlenbachs et al. (2003). This figure makes it seem like the upper part of the crust is in fact sensitive to changing seawater δ18O, so couldn't it actually be used as a proxy for seawater δ18O?"

Response:

The sensitivity to seawater $\delta^{18}O$ is higher at shallow depths of oceanic crust compared to that in the deeper sections as suggested by the reviewer. However, the sensitivity to seawater $\delta^{18}O$ is still significantly smaller than previously assumed. Given the general weak coupling between oceanic crust and seawater $\delta^{18}O$, one has to evaluate the alteration conditions (e.g., spreading rate and permeability) more carefully because they might affect solid rock $\delta^{18}O$ distributions possibly more than seawater $\delta^{18}O$ as discussed in Section 4.1. As a conclusion, I suggested that ophiolites may be interpreted to indicate the insensitivity of oceanic rocks to seawater $\delta^{18}O$ rather than a constant seawater $\delta^{18}O$. One can still use the model to reconstruct seawater $\delta^{18}O$ but the uncertainty would be larger compared to those when using other models in the previous studies that assume strong coupling between oceanic crust and seawater $\delta^{18}O$.

I added a section to Supplementary material where I compared the model simulations that assume 0 ‰ of seawater $\delta^{18}O$ with the modern oceanic crust and Phanerozoic ophiolites including the ophiolite that is studied by Muehlenbachs et al. (2003) to further assess the validity of the present model (Section S2 in Supplementary material). Most of data are comparable to the present simulations. However, the data by Muehlenbachs et al. (2003) is an exception, i.e., their data is significantly smaller than the model prediction. This discrepancy may be attributed to the lower seawater $\delta^{18}O$ during the Paleozoic (Galili et al., 2019) but could also be caused by changes in the permeability and/or rate constant for oxygen isotope exchange (also related to reactive surface area). Please see Section S2 in Supplementary material for more details about the model-data comparison.

In the above comparison with datasets (Section S2 in Supplementary material), I excluded Precambrian ophiolites (including the ophiolite studied by Holmden and Muehlenbachs, 1993) because seawater $\delta^{18}O$ in the Precambrian could have been more significantly deviated from the present-day value (e.g., Galili et al., 2019; Johnson and Wing, 2020) and thus Precambrian ophiolites

are not suited for assessing the validity of the model. Also, Holmden and Muehlenbachs (1993) did not provide explicit depth information of rock samples and thus their data cannot be directly compared with the simulation results.

Changes in manuscript (Page numbers/Line numbers):

I added Section S2 to Supplementary material where I compare the model simulations with more $\delta^{18}O$ datasets from the modern oceanic crust and Phanerozoic ophiolites (P1/L6-P2/L62 in Supplementary material). The added section is referred to in the main text where relevant (P7/L195).

More explanations were added to Section 4 regarding the uncertainty in reconstruction of ancient seawater $\delta^{18}O$ (P12/L361-363).

Specific comment 1:

"Paragraph lines 26-42: In addition, lower temperatures are supported by O-isotopes in phosphates (Blake et al., 2010 Phosphate oxygen isotopic evidence for a temperate and biologically active Archaean ocean), so it's not just sporadic glacial activity. There are also GCM studies that support non-super hot conditions (Wolf and Toon, 2014, Controls on the Archean Climate System investigated with a global climate model)"

Response:

Blake et al. (2010) suggested temperate climate based on phosphate $\delta^{18}O$ from the Archean sediment. Their finding of the relatively high phosphate $\delta^{18}O$ is in contrast to the trend of phosphate oxygen isotopes reported by Karhu and Epstein (1986, Geochim. Cosmochim. Acta 50, 1745), which should be mentioned when introducing the general sedimentary $\delta^{18}O$ trend (Section 1).

Wolf and Toon (2014) simulated Archean climate under various $CH_4$ and $CO_2$ conditions. However, one cannot conclude whether hot conditions could have been possible or not in the Archean only with the study by Wolf and Toon (2014) because it depends on the constraints on atmospheric $CO_2$ and $CH_4$. More recently, Charnay et al. (2017, Earth Planet Sci. Lett. 474, 97) showed that hot climate in the Archean can be realized in a GCM if weathering feedback is not effective.

Changes in manuscript (Page numbers/Line numbers):

I included the reference of Blake et al. (2010) in the revised manuscript (P2/L28-29), but did not refer to climate models (please see my response above).

Specific comment 2:

"In this same paragraph, it's important to note that the samples from the new Galili et al. study are all

from the Proterozoic and younger, and do not give additional information on the Archean."

Response:
Agreed.

Changes in manuscript (Page numbers/Line numbers):
I revised the relevant sentence to be clearer (P2/L38).

Specific comment 3:
"Line 105-106: is 104 years sufficient? Many low-temperature systems last much longer than this, with additional water circulation"

Response:

$3 \times 10^4$ yr is the duration of time in each iteration, but not the total time duration of hydrothermal simulations. Iterations are repeated $10^3$ times so the total duration of hydrothermal simulation is $3 \times 10^7$ years. This time scale is sufficient to reach the system's steady state as reported in other studies (e.g., Cherkaoui et al., 2003).

Changes in manuscript (Page numbers/Line numbers):
I added more explanations (P4/L121).

Specific comment 4:
"Equation 7: This seems to be a key part of your conclusions, that slow kinetics limit Oisotope buffering. Your constant, 10-8.5, is lower than previous estimates. This value needs a bit more justification. What is the reasoning that field kinetics are slower? Is it just harder to measure?"

Response:
The slower reaction kinetics in the field than in the laboratory has long been recognized and discussed especially regarding mineral dissolution/precipitation (e.g., Pačes, 1983, Geochim. Cosmochim. Acta 47, 1855; Velbel, 1993, Chem. Geol. 105, 89; White and Brantley, 2003; Maher et al., 2004, 2009). However, the cause of the kinetic discrepancy has not been fully understood. Suggested mechanisms include a decline in the reactive surface area with rock age and significantly different porewater residence time and porewater chemistry in the field (e.g., White and Brantley, 2003; Maher et al., 2009). Thus, it is appropriate to account for the kinetic discrepancy in a reactive transport model, as done in other models (e.g., Fantle and DePaolo, 2006, Geochim. Cosmochim. Acta 70, 3883; Moore et al., 2012, Geochim. Cosmochim. Acta 93, 235; Yokota et al., 2013,

Geochim. Cosmochim. Acta 117, 332). Cathles (1983) also used a factor to lower the kinetic rate constant for oxygen isotope exchange down to $10^{-4}$. He indicated that such a factor is necessary to better explain observed oxygen isotope profiles, which is not inconsistent with the present study.

In addition, sensitivity analysis where the rate constant for oxygen isotope exchange at reference temperature (5 °C) is varied from the laboratory value to the reduced value by a factor of $10^4$ (Section S4 in Supplementary material) suggests that the general results and conclusions are not affected by the variations in the rate constant, although the model reproduces the observations best with the standard value, i.e., $10^{-8.5}$ $mol^{-1}$ $kg$ $yr^{-1}$. Please also find that the standard value is not a random number but the geometric mean of the range that could be observed in the field, i.e., from the laboratory value to the reduced value by a factor of $10^4$ that accounts for the field-laboratory discrepancy (a factor of up to $10^3$; e.g., White and Brantley, 2003) and the uncertainty in reactive surface area (a factor of up to 10; Nielson and Fisk, 2010).

Changes in manuscript (Page numbers/Line numbers):
I added explanations on how the standard value of the rate constant for oxygen isotope exchange at reference temperature (5 °C) is determined in Section 2.2 (P5/L147-151), and in Section S4 of Supplementary material (P3/L83-91 in Supplementary material).

I added a section to Supplementary material (Section S6) where I explain the kinetic discrepancy between the laboratory and field and its potential mechanisms in more detail (P5/L144-P6/L181 in Supplementary material).

Specific comment 5:
"In addition, the related material in the supplement (Fig. S7), appears to show a pretty different pattern of δ18O in the crust depending on kex. Can you provide some additional justification?"

Response:
Please see my response to specific comment 4 by Referee #1 where I addressed the issue.

Changes in manuscript (Page numbers/Line numbers):
Please see my changes in manuscript in response to specific comment 4 by Referee #1.

Specific comment 6:
"I grant that your model fits the Oman ophiolite data well, but we know that the δ18O of seawater at the time this formed is not different than today, so perhaps testing your model in a system that we know has a different δ18O value, such as a freshwater system, might be insightful."

Response:

Freshwater systems can be characterized by a relatively-short-term intrusion and later cooling, i.e., little effect of solid rock transport via spreading (e.g., Norton and Taylor, 1979). In such a case without solid rock transport, transient simulation is necessary (e.g., DePaolo, 2006), which cannot be conducted by the present reactive transport model of oxygen isotopes, which simulates only steady state distributions of solid-rock and porewater $\delta^{18}O$ (Section 2.2). Therefore, the present model cannot be applied to freshwater systems. Please also see my response to specific comment 10 by Referee #1 on the related issue.

Instead, in response to the comment, I compared the model simulations with more datasets of the modern oceanic crust and Phanerozoic ophiolites (Section S2 in Supplementary material) to further assess the validity of the model. The model and data are mostly comparable and effects of spreading rate are well predicted by the present model, and the validity of the model is further supported.

Changes in manuscript (Page numbers/Line numbers):

I added Section S2 to Supplementary material where the present simulations are compared with more datasets of oceanic crust and ophiolites (P1/L6-P2/L62 in Supplementary material). This specific section is referred to in the main text where relevant (P7/L195).

Specific comment 7:

"Equation 8: why use this equation for andesite? You say it's similar to Cole et al. for basalt, so what is the advantage?"

Response:

Data by Cole et al. (1987) is limited to the temperature range of their experiments (i.e., 300 to 500 °C). To extrapolate the data by Cole et al. (1987) over the wider temperature range considered in the present study (from 2 to 1200 °C), theoretical models are useful. I adopted the model by Zhao and Zheng (2003) because their model can be applied to the above wide range of temperature and their model for andesite also predicts fractionation factors that are similar to those reported by Cole et al. (1987).

Changes in manuscript (Page numbers/Line numbers):

I revised the relevant sentence to be clearer (P6/L157).

Specific comment 8:

"Why is permeability set to 0 below 6km?"

Response:

6 km depth from the crust/ocean interface is assumed to be the location of the boundary between the oceanic crust and mantle, below which the permeability can be assumed to be significantly reduced as in Cherkaoui et al. (2003). The assumption is also consistent with no water flux at the lower boundary (at 5 km depth) for a smaller calculation domain ($5 \times 5$ km$^2$) by Cathles (1983) (please also see my response to specific comment 10 by Referee #1).

Changes in manuscript (Page numbers/Line numbers):
I added more explanations to the relevant sentence (P4/L98-99).

Specific comment 9:
"Depaolo (2006), which you do cite, found that equilibrium exchange is a good approximation as long as fractures are ~1-4 m apart, as in MOR. Why does your work differ here?"

Response:

DePaolo (2006) adopted a dual porosity model where fractures and rock matrix between fractures are separately treated and argued that isotopic composition of pore fluid can change depending on the reaction length in the rock matrix and fracture spacing. In turn, applying the model to pore fluid data for O and Sr isotopes, DePaolo (2006) suggested fracture spacing can be 1-4 m, although the author also indicated that this estimate may change once the effect of solid rock transport is included. In the above fracture spacing estimate, the temperature of pore fluid and thus the reaction rate are assumed. In other words, equilibrium exchange is not a consequence of 1-4 m fracture spacing, but just an assumption. Thus, fracture spacing of 1-4 m does not necessarily mean that the system must be in equilibrium.

Please also find that I do not argue that isotope exchange equilibrium is not achieved in MOR; in contrary, sections where temperatures are high are characterized by isotope exchange equilibrium (e.g., Fig. 5). The differences of my model from DePaolo's model (2006) include that my model explicitly includes the effect of solid rock transport, which supplies significant O isotopes to the system and buffers porewater $\delta^{18}$O. As discussed in Section 4.3, because of the lack of solid rock transport, DePaolo's model (2006) might overestimate the contribution of seawater to formation of porewater $\delta^{18}$O and tends to assume a relatively strong coupling between solid rock and seawater $\delta^{18}$O as in other models (e.g., Taylor, 1977; Criss et al., 1987; Gregory et al., 1989).

Please also see my response to general comment 1 by Referee #1 on the issue about fracture/crack treatment in the model.

Changes in manuscript (Page numbers/Line numbers):

I revised the manuscript so that it becomes clearer that equilibrium can be achieved in high-temperature sections (P8/L226-227, P8/L242, P8/L243).

Specific comment 10:
"Another study using a similar approach is Cathles, L. M. in The Kuroko and Related Volcanogenic Massive Sulfide Deposits Vol. 5 (eds Ohmoto, H. & Skinner, B. J.) 439–487 (Economic Geology Publishing, 1983). How does your work compare to theirs, which is very similar in approach?"

Response:
The hydrothermal circulation model by Cathles (1983) is similar to my model in that both models are based on energy, mass and momentum conservation. However, Cathles's model is more limited with respect to spatial resolution ($39\times29$ grid with $35\times20$ to $520\times260$ $m^2$ grid cell sizes in Cathles's model (1983) vs. $320\times200$ grid with $1.1\times0.17$ to $330\times82$ $m^2$ grid cell sizes in the present model), the size of calculation domain ($5\times5$ vs. $12\times30$ $km^2$), permeability distribution (assumed permeability that changes with temperature vs. constrained permeability based on observations by Fisher, 1998) and the time scale ($< 50000$ years vs. steady state (reached by $\leq3\times10^7$ years simulations)). Accordingly, experimental setup by Cathles (1983) is suited to simulate effects of local and short-term episodic intrusion and later cooling of oceanic crust while the present model is suited to describing a hydrothermal system that operates on long term.

Oxygen isotope modeling for porewater is similar to that in this study, but not clear for the simulation of oxygen isotopes in solid rock as I could not find the governing equation for solid-rock $\delta^{18}O$ in Cathles (1983). Thus, the importance of $\delta^{18}O$ buffering by transported oceanic rocks cannot be inferred from the work by Cathles (1983). Nonetheless, Cathles (1983) described that the positive and negative $\delta^{18}O$ anomaly relative to the fresh rock $\delta^{18}O$ is caused by non-equilibrium alteration and $\delta^{18}O$ supply from solid rock, respectively, which is consistent with the present simulations (e.g., Fig. 5).

Changes in manuscript (Page numbers/Line numbers):
I added the reference of Cathles (1983) to the revised manuscript and added more explanations to description of the present model (P5/L130, P5/L151-152, P8/L233-236).

Technical comment 1:
"Typo in line 33? Should this be 70-85 degrees C?"

Response:
I thank the reviewer for pointing out the typo. It meant $70\pm15$ °C.

Changes in manuscript (Page numbers/Line numbers):
I corrected the typo (P2/L34).

---

## Author Comment (AC2) · 10 Jun 2020

**Response to Referee #2 (Dr. Itay Halevy)**

I express my gratitude to Dr. Itay Halevy for his useful comments. My response to the reviewer's comments and the corresponding revision are described in detail below. The numbers of pages, lines, equations, tables and figures are those in the revised manuscript unless otherwise described.

Major comment 1:
"Firstly, where the model is validated against ophiolite d18O profiles or estimates of 18O fluxes to/from the oceanic crust, these consistency tests only have meaning when the age (i.e., alteration duration) of the oceanic crust is known, and when the model results at that specific alteration duration are compared to the observations. The model needs to be validated against profiles with better-constrained duration of alteration, perhaps from ODP boreholes."

Response:
Oxygen isotopic data from the modern oceanic crust with known ages including ODP boreholes has suggested that significant oxygen isotope exchange during oceanic crust alteration is completed within the first <10 million years from the ridge axis (Muehlenbachs, 1979) and not recognized afterwards (e.g., Muehlenbachs, 1979; Barrett and Friedrichsen, 1982; Alt and Bach, 2006). On the other hand, the simulations in the main text assume 30 km for the maximum reaction distance from the ridge axis with variable spreading rates from $1\times10^{-2}$ to $30\times10^{-2}$ m yr$^{-1}$, i.e., the time duration for significant oxygen isotope exchange is assumed to be in the range from 0.1 to 3 million years. The assumed range of the time duration (i.e., 0.1 to 3 million years) satisfies <10 million years and therefore is consistent with the constraint from the observations of the modern oceanic crust with known ages including ODP boreholes (Muehlenbachs, 1979). Also, the modern oceanic crustal $\delta^{18}O$ data cannot be used to further constrain or test the time duration for significant oxygen isotope exchange because it is derived from rocks whose ages are mostly > 3 million years (Muehlenbachs, 1979; please also see Table S1 in Supplementary material). Nonetheless, comparison of fluxes as well as oxygen isotope distributions between the simulations and observations is justifiable, given that observed $^{18}O$ fluxes and distributions reported in the literature have been obtained from systems where significant oxygen isotope exchange is completed (e.g., Holland, 1984; Muehlenbachs, 1998).

The mechanisms to explain why effective oxygen isotope exchange ceases at <10 million years from the ridge axis are important to consider the validity of the model, given that the only maximum time duration (i.e., 10 million years) has been derived from the observations (Muehlenbachs, 1979). Possible mechanisms can include a decline in the reaction rate with age often observed in natural systems (e.g., White and Brantley, 2003; Maher et al., 2004). To further assess the plausibility of the assumed time duration (0.1 to 3 million years), I ran an additional numerical experiment that assumes 300 km maximum reaction distance (or 10 million years with the standard spreading rate of $3\times10^{-2}$ m

yr$^{-1}$) from the ridge axis, includes off-axis water flows and implements a decline in the kinetic constant for oxygen isotope exchange with age that is consistent with field and laboratory observations for mineral dissolution by Maher et al. (2004). The simulation indeed showed that significant changes in solid rock $\delta^{18}O$ (e.g., > 2 ‰) with age are no longer recognized at > ~0.1 to 1 million years from the ridge axis (Section S6 in Supplementary material), and thus further confirmed that the assumed time duration for oxygen isotope exchange in the simulations in the main text (0.1 to 3 million years) is reasonable. Please see Section S6 in Supplementary material for further details.

Even though the modern oceanic crustal $\delta^{18}O$ has given only the constraint of <10 million years for the time duration for oxygen isotope exchange (please see above), it can be compared with the present model simulations to further assess the validity of the model, especially regarding the effect of spreading rate on the distribution of oxygen isotopes. Accordingly, I added a section to Supplementary material (Section S2) where I compare the simulations that assume various spreading rates and 0 ‰ for seawater $\delta^{18}O$ with the $\delta^{18}O$ datasets from the modern oceanic crust (including ODP boreholes) and Phanerozoic ophiolites. The comparison suggests that the model can predict the relationships between oceanic rock $\delta^{18}O$ distributions and the spreading rate that are consistent with the observations and thus further supports the validity of the model. Only exceptions are the data by Barrett and Friedrichsen (1982) and Muehlenbachs et al. (2003). The data by Barrett and Friedrichsen (1982) is slightly smaller than the model prediction but can still be explained by the model if a smaller permeability or kinetic constant for oxygen isotope exchange than in the standard parameterization is assumed. The data by Muehlenbachs et al. (2003) from a Paleozoic ophiolite shows solid rock $\delta^{18}O$ that is smaller than the model prediction, which can be attributed to the lower contemporaneous seawater $\delta^{18}O$ (e.g., Galili et al., 2019) as well as a smaller permeability and/or rate constant for oxygen isotope exchange. Please see Section S2 in Supplementary material for the details.

Changes in manuscript (Page numbers/Line numbers):
I added Section S2 to Supplementary material where the model simulations are compared with more datasets of oceanic rock $\delta^{18}O$ available in the literature (P1/L6-P2/L62 in Supplementary material). The added section is referred to in the main text where relevant (P7/L195).

I added Section S6 to Supplementary material where I discuss the plausible range of the time duration for significant oxygen isotope exchange, showing results from an additional numerical experiment that assumes 300 km maximum reaction distance (or 10 million years) from the ridge axis, includes off-axis water flows and implements a decline in the kinetic constant for oxygen isotope exchange with age that is consistent with observations by Maher et al. (2004) (P5/L144-P6/L181 in Supplementary material).

I added more model explanations regarding the calculation domain width and the time duration for significant oxygen isotope exchange referring to the above sections of Supplementary material (P5/L125-128, P9/L276-278, P10/L287-289).

Major comment 2:

"Secondly, all of the insights gained from the model are based on simulation of circulation and O isotope exchange out to a distance of 30 km from the ridge axis, and a more limited investigation of off-axis alteration out to 300 km. The claims made on the basis of these simulations have far-reaching implications. In my opinion, an effort should be made to show that the weak buffering intensity revealed by the model is not an outcome of this limited model spatial domain. In other words, if one considers sustained low-T alteration as the crust continues aging and until it is subducted, do the main findings of this study hold? Are O isotope fluxes still insensitive to seawater d18O? I urge the author to test this, which will provide confidence in the findings."

Response:

As described in my response to major comment 1 by Referee #2, it has been observed that oceanic crust alteration has only limited influences on oxygen isotopes of oceanic rocks after <10 million years from the ridge axis (e.g., Muehlenbachs, 1979; Barrett and Friedrichsen, 1982; Alt and Bach, 2006). Thus, 30 km reaction distance (or 0.1 to 3 million years) from the ridge axis in the simulations in the main text is reasonable. In addition, simulations that assume 300 km reaction distance from the ridge axis show essentially the same results as those from the simulations with 30 km reaction distance from the ridge axis, with respect to the sensitivity of $^{18}O$ flux and oceanic rock $^{18}O/^{16}O$ fractionation to seawater $\delta^{18}O$ (Sections S5 and S6 in Supplementary material). Furthermore, an additional numerical experiment that assumes 300 km reaction distance from the ridge axis, incudes off-axis water flows and implements the decline in reaction kinetics with age further supports that 30 km is wide enough to simulate oxygen isotope exchange during hydrothermal alteration (Section S6 in Supplementary material; please also see my response to major comment 1 by Referee #2). Therefore, the present study's findings will remain valid even in a wider calculation domain.

    The mechanisms to cause the weak buffering in the present simulations are already discussed; oxygen isotope exchange is kinetically prevented from reaching equilibrium in the low temperature sections and oxygen isotopes of deeper solid rocks are buffered by solid rocks transported via spreading rather than circulating seawater. The two mechanisms make oceanic rocks partially decoupled from seawater with respect to oxygen isotopes, resulting in a relatively weak seawater-$\delta^{18}O$ buffering. These mechanisms operate in systems where significant oxygen isotope exchange continue over longer time scales than assumed in the main text, as confirmed by supplementary simulations in Sections S5 and S6 of Supplementary material.

Changes in manuscript (Page numbers/Line numbers):
Please see my changes in manuscript in response to major comment 1 by Referee #2.
    I added more explanations of the mechanisms to cause the partial decoupling between oceanic crust and seawater $\delta^{18}O$ under different spreading rate conditions in Section 3 (P9/L276-279,

P10/L284-289).

Suggestion related to major comment 2:
"Finally, not a concern so much as a suggestion, related to my second major comment. If this detailed modeling reveals an insensitivity to seawater d18O even out to thousands of km from the ridge axis, but a dependence of subducted crust d18O on physical parameters such as the spreading rate and the thickness of sediment draped on the oceanic crust, then it may provide an explanation not only for the invariant d18O of ophiolites, but also for the long-term secular evolution of seawater d18O. Perhaps this is beyond the scope of the current study, but it would be a welcome and timely contribution."

Response:
Please see my response to major comment 2 by Referee #2 on the issue about the time duration for oxygen isotope exchange or the calculation domain width. I do not consider that simulations with thousands of km from the ridge axis is necessary or reasonable, especially when the apparent cessation of significant oxygen isotope exchange at < 10 million years from the ridge axis has been observed and can be explained/simulated with a decline in efficiency of oxygen isotope exchange with age (Section S6 in Supplementary material).

I agree with the reviewer that revealing the long-term control on oxygen isotopic composition of seawater will make a timely contribution. However, the buffering from hydrothermal systems could have been weak as suggested in the present study and thus understanding oxygen isotope exchange through continental weathering could have been more important than previously assumed, which requires additional modeling work, as discussed in Section 4.2. In other words, it could lead to a false conclusion if one discusses the control of oxygen isotopic composition of ancient oceans only based on hydrothermal alteration of oceanic crust, whose contribution to the oceanic $^{18}O$ budget might have been overwhelmed by that from continental weathering in the deep past.

Changes in manuscript (Page numbers/Line numbers):
I modified the relevant sentence in Section 4 to be clearer about the importance of modeling continental weathering to elucidate the control on oxygen isotopes in the ancient oceans (P11/L327-329).

Specific comment 1:
"L19-21: The sentence in these lines can be worded more clearly."

Response:
Agreed.

Changes in manuscript (Page numbers/Line numbers):
I revised the sentence (P1/L19-21).

Specific comment 2:
'L26: It may be worth mentioning that by "authigenic sedimentary rocks" you mean d18O records in carbonate rocks, cherts, phosphorites, glauconites and shales, all of which show a pronounced increase in d18O over Earth history.'

Response:
I could not find the literature which shows a pronounced increase of $\delta^{18}O$ in glauconites over Earth history. Otherwise I agree.

Changes in manuscript (Page numbers/Line numbers):
I added '(e.g., carbonates, cherts, phosphorites and shales)' to the relevant sentence (P2/L26).

Specific comment 3:
"L32: Is the range 70–15°C correct? Shouldn't the second number be larger than 70?"

Response:
I thank the reviewer for pointing out the typo. It meant 70±15 °C.

Changes in manuscript (Page numbers/Line numbers):
I corrected the typo (P2/L34).

Specific comment 4:
'L38: Perhaps "weak" instead of "little"?'

Response:
Agreed.

Changes in manuscript (Page numbers/Line numbers):
Corrected as suggested (P2/L39).

Specific comment 5:

"L38-42: The motivation for reconciling the sedimentary and ophiolite records is more than just being able to use sedimentary d18O records to reconstruct temperatures. The evolution of seawater d18O is driven by the same processes that govern the chemical fluxes to the ocean (e.g., from low-T continental weathering, from hydrothermal alteration of the oceanic crust at both high and low T), with implications for the evolution of ocean chemistry, the attendant productivity of the biosphere and the composition of the atmosphere."

Response:

I agree with the reviewer that oxygen isotopic composition of seawater can be related to the relative magnitude of low- and high-temperature alteration processes, which can further be linked to the evolution of biosphere, atmosphere and hydrosphere, as well as tectonics (e.g., Verard and Veizer, 2019).

Changes in manuscript (Page numbers/Line numbers):
I revised the relevant sentence to be clearer (P2/L42).

Specific comment 6:
"L52-54: The statement in this sentence is not entirely correct. Any mechanism to lower the T of oceanic crust alteration will result in greater enrichment of the altered crust in 18O (and greater removal of this 18O from the ocean, as the author mentions). For a given amount of alteration, the resulting altered crust will be more strongly offset in d18O from the altering fluid (~seawater). In this case, more 18O-depleted seawater gives rise to correspondingly 18O-depleted authigenic minerals, as observed. This same 18O-depleted seawater could concurrently give rise to altered oceanic crust with d18O similar to modern altered oceanic crust, if the alteration T was lower and the mineral-water O isotope fractionation larger. So at least some of these mechanisms may also explain approximately invariant ophiolite d18O."

Response:

The mechanisms to lower seawater $\delta^{18}O$ might have lowered the surface temperature, as inferred from the comment. However, one can still assume that the temperatures of deep sections of ophiolites were high and little affected by surface temperature variations. In such high temperature sections, fractionation factors are small and thus if porewater is equivalent to seawater with respect to oxygen isotopes, the deep sections could directly record seawater $\delta^{18}O$ (though not supported by this study; please see the next paragraph). Thus, ophiolite records, especially those from deep high-temperature sections, have been interpreted to suggest invariant seawater $\delta^{18}O$. This is well illustrated in Fig. 2 of Holmden and Muehlenbachs (1993). Therefore, the mechanisms to lower seawater $\delta^{18}O$ cannot explain age-invariant ophiolite $\delta^{18}O$ records especially those from deep sections.

Please find that the present simulations revealed that the isotopic equivalence between porewater and seawater mentioned above is unlikely in deep sections of oceanic crust because of significant $\delta^{18}O$ buffering via transported solid rocks. Combined with kinetic inhibition in shallower low-temperature sections, oceanic crust is partially decoupled from seawater $\delta^{18}O$. Thus age-invariant ophiolite $\delta^{18}O$ records may be alternatively interpreted to suggest the relative insensitivity of oceanic crust to seawater $\delta^{18}O$.

Changes in manuscript (Page numbers/Line numbers):
I revised the relevant sentence to be clearer (P2/L56).

Specific comment 7:
'L67: "The present study has been undertaken to present…". Suggest rewording.'

Response:
Agreed.

Changes in manuscript (Page numbers/Line numbers):
I revised the sentence (P3/L69-70).

Specific comment 8:
'L95: Should be "length scale"? Also, perhaps "for an e-fold increase"?'

Response:
Agreed.

Changes in manuscript (Page numbers/Line numbers):
Corrected as suggested (P4/L97).

Specific comment 9:
"L107: Why does the model grid extend to 12 km if the rocks are taken to be impermeable below 6 km?"

Response:
I adopted 12 km rather than 6 km to facilitate changes in the location of the crust/mantle boundary (although I did not change the crust/mantle boundary from 6 km in this study).

Changes in manuscript (Page numbers/Line numbers):
I added more explanations to the relevant sentences (P4/L98-99, P5/L124-125).

Specific comment 10 (related to major comments):
"L107: The choice of a domain length of 30 km from the ridge axis has implications for the timescale of the simulation. At the spreading rates investigated in this study (1e-2 to 30e-2 m/y), formation of 30 km (3e4 m) of new crust takes between 1e5 and 3e6 years. How long does it take to reach a steady state for the q-P-T fields over the domain? Presumably much less than the time that it takes newly produced crust to exit the model domain?"

Response:
The simulations of **q**-*P*-*T* and oxygen isotopes calculate only steady-state profiles. Therefore, time to reach steady states is not calculated, though within 30 million years for **q**-*P*-*T* simulations, and the values reported for **q**, *P*, *T* and $\delta^{18}O$ at any distance (or age) from the ridge axis and any depth from the crust/ocean interface do not change with time. Unless there are multiple steady states, simulating transient states will not affect the steady state results. Simulating only steady-state results is justifiable because oxygen isotopic composition of seawater can change only on a long time scale, e.g., $\sim 0.5 \times 10^8$ yr (Holland, 1984), where long-term buffering intensity should be important.

Please also see my response to major comment 2 by Referee #2 on the issue about the assumed calculation domain width.

Changes in manuscript (Page numbers/Line numbers):
I added more explanations on why only steady state is simulated in the present study (P5/L122-124).

Please also see my changes in manuscript in response to major comment 2 by Referee #2

Specific comment 11:
"L112-113: When you refer to the bottom and right boundaries as insulating, do you mean that there is no temperature gradient across these boundaries? Do the results change if you relax this assumption (e.g., using Neumann boundary conditions with a non-zero flux)? You mention what happens when you relax the assumption of impermeability of these boundaries, and it would be good to also mention what happens when you don't assume the boundaries to be insulating."

Response:
Insulating at a given boundary means there is no temperature gradient and thus no heat flux across the boundary, which is reasonable at the right boundary of a wide calculation domain as in Iyer et al.

(2010). Relaxing this boundary condition (e.g. allowing free heat flow) will not significantly affect the results because a simulation with a wider calculation domain (300 km) yielded similar temperature distributions (cf. Fig. S10 in Supplementary material).

Changes in manuscript (Page numbers/Line numbers):
I added more explanations to the relevant sentence (P4/L112-113).

Specific comment 12:
"L140-143: Could you please better substantiate the choice of a lower-than-lab kex? Is it only due to the smaller specific surface area in the field, or are there other factors, too? In the Supplementary Material, it would be good to show the sensitivity to kex up to the highest lab values (10^-6.6 mol/kg y). This would increase confidence in the low d18O buffering capacity of seafloor alteration suggested in this study."

Response:
The reaction kinetic discrepancy between the laboratory and field has long been recognized and its cause has been discussed but not fully understood. Possible mechanisms include significant difference of reactive surface area and residence time of porewater and porewater chemistry between the field and laboratory (e.g., White and Brantley, 2003; Maher et al., 2009). I extended the range of examined $k_{ex}^{ref}$ value to $10^{-6.5}$–$10^{-10.5}$ mol$^{-1}$ kg yr$^{-1}$ in sensitivity analysis in Section S4 of Supplementary material, which covers the laboratory range ($10^{-6.6}$–$10^{-7.2}$ mol$^{-1}$ kg yr$^{-1}$) as suggested by the reviewer. Variations of $k_{ex}^{ref}$ within the above range do not affect the general results and conclusions (Please see Section S4 in Supplementary material for the details).

Changes in manuscript (Page numbers/Line numbers):
I added more explanation on how the standard $k_{ex}^{ref}$ value is obtained in Section 2.2 (P5/L147-152), and in Section S4 of Supplementary material (P3/L83-91 in Supplementary material).

Also, the range of $k_{ex}^{ref}$ value examined in Section S4 of Supplementary material was extended and associated figures were modified (P3/L92-P4/L102, P17, P18 in Supplementary material).

Specific comment 13:
'L151-152: "The first term on the right-hand side…, while the second term represents the hydrodynamic dispersion.".'

Response:
Agreed.

Changes in manuscript (Page numbers/Line numbers):
Corrected as suggested (P6/L163-164).

Specific comment 14:
"L158: Is the O isotope model insensitive to the assumption of impermeability, like the q-P-T model?"

Response:
When the calculation domain is wide enough, the right boundary can be reasonably assumed to be impermeable with respect to flux of oxygen isotopes via water, given the observation that significant oxygen isotope exchange is limited within < 10 million years from the ridge axis. Simulations with a wider calculation domain show essentially the same results regarding the sensitivity of $^{18}O$ distributions and flux to seawater $\delta^{18}O$ as those by simulations with 30 km calculation domain width (Sections S5 and S6 in Supplementary material). Thus, the calculation domain seems to be wide enough to assume no $^{18}O$ flux via water at the right boundary. Please also see my response to major comment 2 by Referee #2 on the issue about the assumed calculation domain width.

Changes in manuscript (Page numbers/Line numbers):
I referred to Section 2.1 in the relevant sentence where I state that calculation domain is wide enough that changing the right boundary will not have significant influences on the results (P6/L170).

Specific comment 15:
"L163: The Results section contains a lot of discussion. It may be useful to combine the Results and Discussion sections."

Response:
I intended to present the details of results and mechanisms to cause the results in the Results section. Implications of the results and mechanisms from the Results section are discussed in the Discussion section. I consider keeping these two sections separate will be useful to the reader.

Changes in manuscript (Page numbers/Line numbers):
I did not make any specific changes in response to the comment (please see my response above). Please note that I revised the manuscript substantially so that it is easier for the reader to understand the manuscript (please see my changes in manuscript in response to other specific comments by Referee #2).

Specific comment 16:

'L168 and elsewhere: "Ma" is usually reserved for millions of years ago. When referring to millions of years, "Myr" is more commonly used.'

Response:

Agreed.

Changes in manuscript (Page numbers/Line numbers):

I changed Ma to Myr where appropriate (P6/L180, P7/L186, P7/L192, P23).

Specific comment 17:

"L170: The modeled water mass flux is not only within the range of Elderfield and Schultz (1996), it is quite close to their recommended value of 3(±1.5)e13 kg H2O/y."

Response:

Agreed.

Changes in manuscript (Page numbers/Line numbers):

I added a description of the recommended value to the sentence (P7/L182-183).

Specific comment 18 (related to major comments):

"L172-174: There is nothing special about the distance of 30 km from the midocean ridge - if alteration is a sustained process, then there will be some distance at which the model d18O profiles most closely resemble the observations. For a different spreading rate, "consistency with observations" could be reached at a different distance from the spreading center, as suggested by Fig. 8. Consistency can be assessed (or the model calibrated, alternatively) only with independent knowledge on the age of the profiled crust - how long was the sampled crust altered, and does the model resemble the d18O profile in that crust at a comparable duration of alteration. The model should be tested against d18O profiles in crust with a known duration of alteration (perhaps in ODP boreholes)."

"The above relates to a bigger issue, which is the somewhat arbitrary choice of 30 km as the edge of the model domain. Does alteration of the oceanic crust stop farther out from the spreading center? Again, Fig. 8 suggests that this is not the case. In panel (a) of that figure a lower spreading rate results in much more 18Oenriched altered crust than at higher spreading rates (Fig. 4, 8b, 8c). Would this degree of enrichment not be reached farther out from the spreading center at the higher spreading rates? Does the proposed insensitivity to seawater d18O hold if alteration continues over the lifetime of an

oceanic plate?"

"To address this, the author should perform simulations out to much greater distances from the spreading center and identify the distance from the ridge at which the isotopic composition no longer changes. I presume this distance will depend on the model parameters, and this may affect the sensitivity of the ultimate isotopic composition of the crust on seawater d18O. I don't know if this request is practical, given the computational cost of extending the simulation out to thousands of km from the ridge. If not, a way to parameterize the behavior farther away from the ridge with continued water-rock interaction and O isotope exchange should be developed."

"As an aside, constraints on the distance to which water-rock interactions continue to change the isotopic composition of oceanic crust have implications for the effect of oceanic crust alteration on the isotopic composition of seawater. For example, if alteration continues over much of the lifetime of an oceanic plate, then slower seafloor spreading in the Precambrian, as suggested in several recent studies (several papers from Korenaga over the past decade; Fuentes et al., 2019), would lead to subduction of older, more 18O-enriched crust, leaving the ocean 18O-depleted (Galili et al., 2019)."

Response:
Please see my response to major comment 1 by Referee #2 on the issue about comparison with more recent oceanic crust data with known ages (including data from ODP boreholes).

Please see my response to major comment 2 by Referee #2 on the issue about the assumed calculation domain width. Please find that extending the calculation domain width to thousands of km is not reasonable given that observations suggest that significant oxygen isotope exchange is limited within <10 million years from the ridge axis. Instead I conducted an additional simulation to evaluate the reasonable time duration for significant oxygen isotope exchange, which supports that the assumed calculation domain width of 30 km is reasonable (please see Section S6 in Supplementary material and my response to major comment 2 by Referee #2 for the details). Also, simulations with 300 km calculation domain width show essentially the same results regarding the sensitivity of $^{18}$O distributions and flux to seawater $\delta^{18}$O as those in the simulations with 30 km calculation domain width, supporting that the present study's findings are robust.

The spreading rate affects oxygen isotope exchange both at low and high temperatures, and the net flux is affected also by changes in total oxygen supply, as discussed in Section 3.3. The difference caused by changes in the spreading rate is not solely caused by the different time duration but also by different distributions of local water/rock ratio (Fig. 7). This can be confirmed from comparison of Fig. 4 with Figs. S13 and S16 in Supplementary material; the general feature of crustal $\delta^{18}$O, especially that in the deep high-temperature section, is not affected by changes in the calculation domain width if other parameters including the spreading rate are the same. Also, residence time of oceanic crust is not necessarily equivalent to the time duration for significant oxygen isotope exchange (e.g., Muehlenbachs, 1979; Section S6 in Supplementary material). Accordingly, it is not reasonable to discuss changes in the relative significance of low-temperature alteration against high-temperature

alteration only based on the residence time of oceanic crust. Please also see my response to suggestion related to major comment 2 by Referee #2 where I discuss the importance of continental weathering in the oceanic $^{18}$O budget.

There is an uncertainty in the spreading rate during the Precambrian and values can be different between models (e.g., Phipps Morgan, 1998; Korenaga et al., 2017). The present study conducted simulations with a range of spreading rate possible during the Precambrian, and all simulation results show the relative insensitivity of oceanic rocks to seawater $\delta^{18}$O (Section 3.3). Therefore, the main conclusions in the present study are not affected by the uncertainty in the spreading rate during the Precambrian.

Changes in manuscript (Page numbers/Line numbers):
Please see my changes in manuscript in response to major comments 1 and 2 and suggestion related to major comment 2 by Referee #2.

I added more explanations to descriptions of the calculation results with different spreading rates in Section 3.3 (P9/L276-279, P10/L284-289).

Specific comment 19:
"L178-179: The sentence in these lines is difficult to understand. Suggest rephrasing."

Response:
Agreed.

Changes in manuscript (Page numbers/Line numbers):
I revised the sentence (P7/L190-192).

Specific comment 20 (related to major comments):
"L180-189: The model d18O profiles in the bulk rock and the 18O fluxes from high- and low-T alteration are reported in these lines and compared with available observations and previous estimates. As in comment #18, consistency with the observed profiles has meaning only if the model and observed profiles are of an equivalent age (i.e., alteration duration). Likewise, the consistency between model 18O fluxes and previous estimates has meaning only if the estimates were made on the basis of altered crust of a comparable age."

Response:
Please see my response to major comment 1 by Referee #2 where I addressed the issue.

Changes in manuscript (Page numbers/Line numbers):

Please see my changes in manuscript in response to major comment 1 by Referee #2.

Specific comment 21:

"Sections 3.2, 3.3 and onwards: The results, interpretations and implications in the rest of the manuscript should be consistent with the tests performed in response to comments #18 and 20 above."

Response:

Please see my response to specific comments 18 and 20 by Referee #2.

Changes in manuscript (Page numbers/Line numbers):

Please see my changes in manuscript in response to specific comments 18 and 20 by Referee #2.

Specific comment 22 (related to major comments):

"L207-212: As in comments #18 and 20, does the distance from equilibrium keep decreasing past 30 km? If it keeps decreasing, does this affect the proposed insensitivity to seawater d18O?"

Response:

Please see my response to major comments 1 and 2 by Referee #2 where I addressed the issue.

 The distance from equilibrium decreases past 30 km if efficiency of oxygen isotope exchange does not decrease with age (Section S5 in Supplementary material). Observations of oceanic crustal $\delta^{18}O$ with known ages as well as a simulation on a wide calculation domain (300 km) with implementing off-axis flows and decline in efficiency of oxygen isotope exchange with age show apparent cessation of the decrease in the distance from equilibrium around 3-30 km from the ridge axis (Section S6 in Supplementary material).

Changes in manuscript (Page numbers/Line numbers):

Please see my changes in manuscript in response to major comments 1 and 2 by Referee #2.

 I added explanations that enhancement of oxygen isotope exchange in a wide calculation domain in Section S5 of Supplementary material disappears when implementing the decrease of the reaction efficiency with age (Section S6 in Supplementary material) to Section S5 in Supplementary material (P5/L139-142 in Supplementary material).

Specific comment 23:

"L213: Perhaps it would be useful to mention that the reason for the near-equilibrium in the deeper

parts of the section are due to the higher T."

Response:
Agreed.

Changes in manuscript (Page numbers/Line numbers):
Revised as suggested (P8/L226).

Specific comment 24:
'L238: Perhaps change "not inconsistent with" to "consistent with"?'

Response:
Agreed.

Changes in manuscript (Page numbers/Line numbers):
Revised as suggested (P9/L255).

Specific comment 25:
"L235-238: The way these results are reported is very hard to take in, with all of the numbers and parentheses within parentheses. Suggest rewording."

Response:
Agreed.

Changes in manuscript (Page numbers/Line numbers):
I revised the sentence to be clearer (P9/L254-257).

Specific comment 26:
'L252: "spreading" and "weaker" are misspelled.'

Response:
I thank the reviewer for pointing out the misspelled words.

Changes in manuscript (Page numbers/Line numbers):
I corrected the misspelled words (P9/L271).

Specific comment 27:

"L255-278: The two paragraphs in these lines are less well-written than the previous text. Suggest editing for grammar, language and clarity."

Response:

Agreed.

Changes in manuscript (Page numbers/Line numbers):

I revised the two paragraphs to be clearer (P9/L273-P10/L304).

Specific comment 28:

"L273-278: The model of Kasting et al. (2006) included the effect of overburden (ocean depth) on the depth in the crust at which water reached the critical point, leading to changes in the capacity of hydrothermal systems to transport heat and, consequently, on the temperature profile of water-rock interactions. Are such water phase changes considered in the present model, and if not, could that be an additional reason for disagreement with the results of Kasting et al. (2006)? Please discuss."

Response:

Water properties calculated as functions of temperature and pressure are comparable to those by Kasting et al. (2006). Therefore, the different conclusion of this study regarding the effect of water depth stems from the difference in the model configuration. The significant difference of this study's model from Kasting et al. (2006) model includes that the permeability and water properties can change two-dimensionally. Also, Kasting et al. (2006) assume that hydrothermal circulation is represented only by water close to the critical pressures and temperatures to maximize the heat transport and they mentioned the need to verify this assumption in two-dimensional simulations.

Changes in manuscript (Page numbers/Line numbers):

I revised the relevant sentence to be clearer (P10/L302-304).

Specific comment 29 (related to major comments):

"L281-282: This statement needs to be reevaluated following the tests requested in comments #18, 20, 21, 22. Hopefully, it still holds."

Response:

Please see my response to specific comments 18, 20, 21 and 22 by Referee #2.

I confirmed that the statement still holds by conducting an additional experiment where plausible time duration for significant oxygen isotope exchange was examined (Section S6 in Supplementary material).

Changes in manuscript (Page numbers/Line numbers):
Please see my changes in manuscript in response to specific comments 18, 20, 21 and 22 by Referee #2.

Specific comment 30:
'L283-286: This sentence is awkwardly worded. Suggest "By comparison, the simulated solid rock d18O values fall within this range for seawater d18O values $\geq -10$, $-8$ and $-2$‰ at a spreading rate of 1e-2, 3e-2 and $\geq$9e-2 m yr^-1, respectively (Figs. 4, 8)." Related to the above, what are the average Archean/Proterozoic/Phanerozoic seafloor spreading rates suggested in previous studies, and what are the implications for the evolution of the 18O-buffering strength of hydrothermal alteration of oceanic crust over Earth history?'

Response:
I agree to revise the sentence.

Average spreading rates over the Earth history, especially during the Precambrian, are not fully known and diverse values have been suggested (e.g., Phipps Morgan, 1998; Korenaga et al., 2017). This is the reason why I adopted a range of spreading rate that covers the suggested diverse average values. Unless we can constrain the evolution of tectonics, it remains uncertain how [18]O buffering intensity evolved. Despite this uncertainty, the buffering intensity should have been weaker than previously assumed and continental weathering contribution is likely to have been more significant than previously assumed.

Changes in manuscript (Page numbers/Line numbers):
I revised the sentence as suggested (P10/L309-311).

Also, more explanations were added to the sentences on the spreading rates in the Precambrian (P9/L251-253).

Specific comment 31:
"L286-288: There are values of seawater d18O that are inconsistent with the range observed in ophiolites, right? Perhaps mention those values? Related to this, it appears that the model reproduces the range observed in ophiolites irrespective of seawater d18O mostly at low spreading rates. It is

worth mentioning that estimated Precambrian seafloor spreading rates were slower than Phanerozoic rates."

Response:

I consider describing the seawater $\delta^{18}O$ values that are either consistent or inconsistent with ophiolite data is sufficient. I described only the consistent values but not inconsistent values.

Please see my response to specific comment 30 on the issue about the spreading rate. Precambrian spreading rates can be either higher or lower than Phanerozoic spreading rates depending on the model (e.g., Phipps Morgan, 1998; Korenaga et al., 2017).

Changes in manuscript (Page numbers/Line numbers):

No specific changes were made in response to the comment on the inconsistent seawater $\delta^{18}O$ values (please see my response above).

Please see my changes in manuscript in response to specific comment 30 by Referee #2.

Specific comment 32:

"Section 4.3: This section could also benefit from editing for grammar, language and clarity."

Response:

I agree to revise.

Changes in manuscript (Page numbers/Line numbers):

I revised Section 4.3 to be clearer (P11/L341-P12/L363).

Specific comment 33:

"Fig. 1: The labels on contours in panels b and d can be moved and spread out so that they are more easily seen. In panel b, orienting the text sub-parallel to the contours near the bottom and right domain boundaries would work nicely. In panel d, orienting the text sub-parallel to the contours near the left boundary would work."

Response:

I agree to revise the figure.

Changes in manuscript (Page numbers/Line numbers):

Corrected as suggested (P20).

Specific comment 34:
"Maybe it's just on my laptop, but there are fine horizontal and vertical lines on the filled contour plots with a continuous color scale (Fig. 1, 5, 7)."

Response:
Fine horizontal and vertical lines do not appear in my computers (both laptop and desktop). I guess they might appear depending on the settings in the pdf reading software, but not when printed.

Changes in manuscript (Page numbers/Line numbers):
No specific changes were made in response to the comment.

Specific comment 35:
'Fig. 2: Suggest changing "0, –6 and –12 ‰ of seawater d18O" to "at seawater d18O values of 0, –6 and –12 ‰".'

Response:
Agreed.

Changes in manuscript (Page numbers/Line numbers):
Corrected as suggested (P21).

Specific comment 36:
'Fig. 4 caption: "Ma" -> "Myr". Suggest changing "0, –2, …, –12 ‰ of seawater d18O" to "at seawater d18O values of 0, –2, …, –12 ‰". Note that this comparison is meaningful only for crust of a similar alteration duration (see comments #18, 20, 21, 22).'

Response:
Agreed.
        Please see my response to major comment 1 by Referee #2 on the issue about comparison with curst with various ages.

Changes in manuscript (Page numbers/Line numbers):
Corrected as suggested (P23).
        Please see my changes in manuscript in response to major comment 1 by Referee #2.

Specific comment 37:

'Fig. 5 caption: "0 ‰ of seawater d18O" -> "a seawater d18O value of 0 ‰".'

Response:

Agreed.

Changes in manuscript (Page numbers/Line numbers):

Corrected as suggested (P24).

Specific comment 38:

"SM L33: As mentioned in comment #12, the choice of a factor of 10 for the uncertainty is arbitrary. It would be good to perform an additional simulation at kex = $10^{-6.5}$. If the results are indeed insensitive to the value of kex, this will not matter much for the buffering intensity, and it would provide confidence in the proposed insensitivity of seafloor alteration to seawater d18O."

Response:

Please see my response to specific comment 12 by Referee #2.

Changes in manuscript (Page numbers/Line numbers):

Please see my changes in manuscript in response to specific comment 12 by Referee #2.

Specific comment 39:

'Fig. 7 caption: The sentence starting with "Spreading rate" is awkward. Suggest rewording.'

Response:

Agreed.

Changes in manuscript (Page numbers/Line numbers):

I revised the sentence (P26).

Specific comment 40:

'Fig. 8: Suggest decreasing font size of axis tick labels. Also, "0, −2, …, −12 ‰ of seawater d18O" -> "at seawater d18O values of 0, −2, …, −12 ‰".'

Response:

Agreed.

Changes in manuscript (Page numbers/Line numbers):

Revised as suggested (P27).

Specific comment 41:

"SM Section S3, Figs. S7, S8: Looking at Fig. S7, there are significant differences between the profiles at a different value of kex. Please explain mechanistically why the buffering intensity ends up being so similar."

Response:

The buffering intensity is determined by the sensitivity of solid rock and porewater $\delta^{18}O$ to seawater $\delta^{18}O$. Although the absolute values are different with different reference rate constants for oxygen isotope exchange, the sensitivity is little affected, as can be seen from the limited ranges of changes in solid rock $\delta^{18}O$ compared to the imposed range of seawater $\delta^{18}O$. The mechanisms to accomplish weak buffering are described in the main text (kinetic inhibition and $^{18}O$ supply via spreading solid rocks in the shallow and deep sections of oceanic crust, respectively) and the same mechanisms can be applied to the simulations in Section S4. Please find that the section number has been changed from S3 in the previous SM to S4 in the revised SM.

Changes in manuscript (Page numbers/Line numbers):

I added more explanations to Section S4 in Supplementary material (P3/L97-P4/L102 in Supplementary material).

Specific comment 42 (related to major comments):

"SM Section S4: A major concern of any clued reader will be that the current model only extends out to an oceanic crust age of 1e5 to 3e6 years (see many of my comments above). As such, I suggest moving some of this section to the main text, perhaps in the discussion."

Response:

As in my response to major comment 2 by Referee #2, the modern observations of oceanic crust with known ages suggest that significant oxygen isotope exchange is limited within less than 10 million years from the ridge axis and thus the assumed calculation domain width is reasonable. Supplementary experiments in Supplementary material (e.g., experiments in Sections S5 and S6 of Supplementary material) were conducted only to confirm that the assumed time duration for significant oxygen isotope

exchange is reasonable. Moving these supplementary simulations to the main text will distract the reader from the point of this study and thus was avoided for clarity.

Changes in manuscript (Page numbers/Line numbers):
Please see my changes in manuscript in response to major comment 2 by Referee #2.

Specific comment 43 (related to major comments):
"SM Section S4: Is a distance of 300 km from the ridge axis sufficient? Does the model d18O of the crust stop evolving after this distance? As with many of my comments above, it is important to constrain the change in the profiles as the crust ages and run the simulations out to a distance beyond which the additional change is negligible."

Response:
Please see my response to major comments 1 and 2 by Referee #2 where I addressed the issue.

Changes in manuscript (Page numbers/Line numbers):
Please see my changes in manuscript in response to major comments 1 and 2 by Referee #2.

Specific comment 44:
"SM Section S4: The finding that the buffering intensity is no different from the standard case when off-axis alteration is included is very important, and it is understandable that the author focuses on this aspect, given the focus of the paper. However, there is a missed opportunity here, in my opinion, which is an exploration of ways in which changes through Earth history in seafloor spreading rates and oceanic plate lifetimes affect the net budget of 18O. Fig. S13 clearly shows that despite similar buffering intensities, the cases with off-axis circulation differ substantially in the net 18O flux from the standard case. If the proportion of off-axis alteration out of the total alteration has changed through time (e.g., changing spreading rate, changing sediment cover, changing crustal thickness), the current model can help to explain the change in seawater d18O suggested on the basis of the O isotope record in authigenic minerals. Perhaps this is beyond the scope of the current contribution."

Response:
Simulations in Section S5 (S4 in the previous SM) ignore a mechanism that can explain the apparent cessation of oxygen isotope exchange at < 10 million years (Muehlenbachs, 1979) because the specific section only focuses on the effect of off-axis water flows on oxygen isotopic composition of oceanic crust. An additional simulation which further implements a decline in efficiency of oxygen isotope exchange with age in a wide calculation domain (300 km) is closer to the standard simulation where

contributions from the low- and high-temperature alteration are comparable. Accordingly, one should not discuss the control of oxygen isotopes in the ancient ocean by changing the calculation domain width, unless the mechanisms of apparent cessation of oxygen isotope exchange during oceanic crust alteration are fully known, which should be studied in the future work. Please find that the results and conclusions remain valid even with assuming a wide calculation domain (i.e., 300 km) (Sections S5 and S6 in Supplementary material). Please also see my response to suggestion related to major comment 2 by Referee #2 where I discuss the importance of continental weathering in the oceanic $^{18}O$ budget.

Changes in manuscript (Page numbers/Line numbers):
Please see my changes in manuscript in response to specific comment 22 and suggestion related to major comment 2 by Referee #2.

Specific comment 45:
"SM L44-46: Please elaborate on the basis for the notion that the oceanic crust is altered within 10 Myr of its formation. The author's off-axis simulations suggest continued low-T alteration for much longer durations."

Response:
Please see my response to specific comment 44 by Referee #2 where I addressed the issue.

Changes in manuscript (Page numbers/Line numbers):
Please see my changes in manuscript in response to specific comment 44 by Referee #2.
     I added explanations on why simulations in Section S5 of Supplementary material show the enhancement of oxygen isotope exchange at low temperature (P5/L136-139 in Supplementary material).

Specific comment 46:
"SM L51: What is the approximate sediment thickness required for this additional 10 MPa? With a density of 2700 kg/m^3 and an assumed porosity of 0.5, about 550 m of sediment are required. Please comment on the plausibility of this at 300 km from the spreading center (given, e.g., Straume et al., 2019) - to me this seems high. Fisher and Becker applied pressures ≤1-3 MPa, up to an order of magnitude less than here. Is it possible to overcome the numerical issues and perform the off-axis simulations with less of an overburden and lower imposed pressures?"

Response:

As stated in Section S5 of Supplementary material, imposed pressures and sediment burden intervals are larger than suggested by Fisher and Becker (2000) because of limited resolution in the wider calculation domain. One may resolve the problem by adopting finer grid cells, but it will not be practically possible as it can require the calculation time as long as weeks even to months.

Sediment thickness to cause an additional pressure of 10 MPa can calculated to be ~1200 m, with a sediment grain density of 2700 kg m$^{-3}$ and a porosity of 0.5. Please note that pressure caused by water need be excluded from the calculation of additional pressure because water pressure is already included in the default hydrostatic pressure of 25 MPa at the curst/ocean interface.

The sediment thickness to cause the additional pressure of 10 MPa (e.g., 1200 m) is close to the maximum value within 10 million years from the crust formation: sediment thickness can be as thick as ~1 km on crust that is <10 million years old depending on the latitude (Fig. S4 in Müller et al., 2008, Science 319, 1357).

As long as off-axis flows are numerically implemented, sediment thickness does not have to be so realistic because the purpose of the specific section is not to mechanistically explain the off-axis flows as in Fisher and Becker (2000) but to examine the effect of off-axis flows on oxygen isotopes of oceanic crust. Indeed, the implemented off-axis flows satisfy the constraint on the total off-axis water flux from observations as described in Section S5 of Supplementary material. Therefore, even if the model can be improved with respect to off-axis flow simulation, the results and conclusions will remain the same.

Changes in manuscript (Page numbers/Line numbers):
I added more explanations to Section S5 in Supplementary material (P4/L111-117, P4/L127 in Supplementary material).

Specific comment 47:
"SM Section 4 and elsewhere: Please replace "Ma" with "Myr", as necessary (see comment #16)."

Response:
Agreed.

Changes in manuscript (Page numbers/Line numbers):
Revised as suggested (P4/L135 in Supplementary material).

Specific comment 48:
"SM L74-78: See comment #44. There is a missed opportunity here."

Response:

Please see my response to specific comment 44 by Referee #2 where I addressed the issue.

Changes in manuscript (Page numbers/Line numbers):

Please see my changes in manuscript in response to specific comment 44 by Referee #2.

Specific comment 49:

'SM Fig. S1 caption: "0, −6 and −12 ‰ of seawater d18O" is grammatically awkward. I suggest changing this (in two places in the caption) to "at seawater d18O values of 0, −6 and −12 ‰". Likewise, suggest "adopt a spreading rate of R1, R2 and R3, respectively." instead of the current text.'

Response:

Agreed.

Changes in manuscript (Page numbers/Line numbers):

Revised as suggested (P11 in Supplementary material).

Specific comment 50:

"SM Fig. S2 caption: Same as comment #49. This wording appears also in several of the other SM figures. Suggest changing."

Response:

Agreed.

Changes in manuscript (Page numbers/Line numbers):

Revised as suggested (P12, P16, P17, P20, P21 in Supplementary material).

Specific comment 51:

"SM Fig. S3, S5, S6, S7, S8, S12, S13: Suggest smaller font size on axis tick labels."

Response:

Agreed.

Changes in manuscript (Page numbers/Line numbers):

Revised as suggested (P14, P16, P17, P18, P21, P22 in Supplementary material).

---

## Author Response (AR2)

Concerning the paper entitled 'Quantifying the buffering of oceanic oxygen isotopes at ancient midocean ridges' (se-2019-169)

June 29, 2020

Dear Editors Drs. Patrice Rey and Joachim Gottsmann,

I express my gratitude to you for considering and accepting my paper for publication in SE. According to Topical Editor's comments, I made the model codes available to public and have assigned a DOI to the version used for this paper. Corresponding revision was made in Code availability section. Please see just below for the list of all relevant changes in the revised manuscript and further below for the marked-up manuscript.

List of all relevant changes

[revised manuscript text omitted]